# GLC_FCS10: a global 10-m land-cover dataset with a fine classification system from Sentinel-1 and Sentinel-2 time-series data in Google Earth Engine

Xiao Zhang [1,2], Liangyun Liu [1,2,3*], Tingting Zhao [1,4], Wenhan Zhang [1,2,3], Linlin Guan[1,2], Ming Bai [1,5], and Xidong Chen [6]

International Research Center of Big Data for Sustainable Development Goals, Beijing, 100094, China.
Key Laboratory of Digital earth Science, Aerospace information Research institute, Chinese Academy of Sciences, Beijing, 100094, China.
School of electronic, electrical and communication engineering, University of Chinese Academy of Sciences, Beijing, 100049, China.
School of Geography and Ocean Science, Nanjing University, Nanjing, Jiangsu 210023, China.
College of Geomatics, Xi'an University of Science and Technology, Xi'an 710054, China.
Future Urbanity & Sustainable environment (FUSE) Lab, the University of Hong Kong, Hong Kong, 999007, China.

Corresponding author: Liangyun Liu (liuly@radi.ac.cn)

## Abstract

The continuous development of remote sensing techniques provides ample opportunities for high-resolution land-cover mapping. Although global 10-m land-cover products have made considerable progress over past few years, their simple classification system makes it difficult to meet the needs of diverse applications. In this work, we propose a hierarchical land-cover mapping framework to produce a novel global 10-m land-cover dataset with a fine classification system (called GLC_FCS10) using Sentinel-1 and Sentinel-2 time-series observations in 2023. First, the globally distributed training samples are hierarchically obtained from multisource prior products after applying a series of refinements. Then, a combination of hierarchical land-cover mapping, local adaptive modeling, and multisource features is used to produce land-cover maps for each $5 \times 5$ geographical tile. Next, using 56121 globally distributed validation samples and a third-party validation dataset (LCMAP_Val), the GLC_FCS10 is assessed. The GLC_FCS10 achieves an overall accuracy of 83.16% and a kappa coefficient of 0.789 globally and an overall accuracy of 85.09% in the United States. Meanwhile, comparisons with five released 10- or 30-m land-cover products also demonstrate that GLC_FCS10 has higher accuracy and captures more diverse land-cover information than three of the released global 10-m land-cover products. In summary, the novel GLC_FCS10 land-cover maps can provide important support for high-resolution land-cover related research and applications. The GLC_FCS10 can be freely access via https://doi.org/10.5281/zenodo.14729665 (Liu and Zhang, 2025).

## 1. Introduction

Land-cover information is a vital component of global climate change research and plays an important role in climate change mitigation, biodiversity protection, and global food security (Foley et al., 2005; Liu et al., 2021). With advancements in satellite techniques and computational and storage capabilities, global land-cover mapping has made substantial progresses. A series of global land-cover products, ranging from 1-km to 10-m resolutions, has been continuously released (Giri et al., 2013; Liu et al., 2021). Recently, Wang et al. (2023) reviewed the characteristics of global land-cover products and found that land-cover mapping has evolved from coarse to high spatial resolution. Currently, four global 10-m land-cover products are available, including FROM_GLC10 (Gong et al., 2019), ESRI LC (Karra et al., 2021), European Space Agency (ESA) WorldCover (Buchhorn et al., 2020), and Dynamic World (Brown et al., 2022). However, all of these products use a simple classification system, which limits their applicability for specific and fine applications (Zhang et al., 2021). Meanwhile, the work of Zhao et al. (2023) explained that FROM_GLC10, ESRI LC, and ESA WorldCover have relatively low consistency and accuracy. Thus, developing an accurate global 10-m land-cover dataset with a fine classification system is still necessary.

The diversity of land-cover types depends on the training samples, and there are two options to generate a globally distributed training pool—visual interpretation and automated derivation from prior products (Zhang and Roy, 2017; Zhang et al., 2021). Visual interpretation means that experts or volunteers interpreted the land-cover information through high-resolution imagery, Google Earth Streetview photos, or other auxiliary datasets. For example, the training samples in FROM_GLC10 were derived from expert interpretations and contained approximately 93,000 sites worldwide (Gong et al., 2013), and the ESA WorldCover used 20 trained experts to collect approximately 141,000 training locations from the Geo-Wiki engagement platform (Buchhorn et al., 2020). Obviously, training sites from the "visual interpretation" option can ensure high quality (Ban et al., 2015), however, the problems of cost and time are not to be ignored. The diversity of the land-cover classification system also relies on the experts' prior knowledges. The "automated derivation from prior products" option generates the training sites from prior land-cover datasets after taking some refinements or validations (Radoux et al., 2014; Zhang et al., 2021; Zhang et al., 2024c). For example, the training areas in the GLC_FCS30 were collected from a combination of time-series MCD43A4 surface reflectance data and CCI_LC land-cover products after using some spatiotemporal purification methods, and these automatically derived training samples supported high-accuracy land-cover mapping with overall accuracy of 82.5% (Zhang et al., 2021). The automated option enables more efficient collection of globally confident training samples; however, the classification errors of prior land-cover products were also easier to transfer into the derived training samples (Zhang and Roy, 2017). Therefore, it is critical to avoid transferring error into the training samples.

Another great challenge in global land-cover mapping lies in the choice of suitable methodologies. Currently, the vast majority of global land-cover mapping ignores the complexity and sparsity of various land-cover types and completes a mapping project with a single classification model (Friedl et al., 2010; Gong et al., 2013), which leads to considerable uncertainties in sparse (e.g., impervious surfaces) or complex (e.g., wetlands and shrubland) land-cover types (Karra et al., 2021; Zhang et al., 2021; Zhao et al., 2023). Several measures were taken to improve the performance of large-area land-cover mapping, such as: local adaptive modeling (Defourny et al., 2018; Li et al., 2023; Zhang and Roy, 2017; Zhang et al., 2021), hierarchical land-cover mapping (Chen et al., 2015; Sulla-Menashe et al., 2019), or integration of multisource datasets (Yu et al., 2014; Zhang et al., 2020).

Specifically, the local adaptive modeling first split study area into many local areas and further trained corresponding classification models within each local region to improve the ability to capture regional characteristics. Zhang and Roy (2017) found that local adaptive modeling had higher accuracy than single global land-cover modeling, however, it also needs enough training samples to support regional modeling. Hierarchical land-cover

mapping divides the land surface into various land-cover layers, and some complicated land-cover layers may need to be treated independently. Taking wetlands as an example, most global land-cover products perform poorly on wetlands because of their varied spectral characteristics and complicated spatiotemporal features (Buchhorn et al., 2020; Gong et al., 2019; Zhang et al., 2021). The GlobeLand30 achieved 74.87% accuracy with wetlands because this land-cover type was treated independently (Chen et al., 2015). Hierarchical land-cover mapping gives more attention to complicated land-cover types by importing more prior knowledge (Chen et al., 2015) or adding sufficient high-confidence training samples (Zhang et al., 2023b). Lastly, many previous works have demonstrated the integration of multisource datasets, such as optical imagery (Landsat and Sentinel-2) and Sentinel-1 single-aperture radar (SAR) data, to improve the identification of impervious surfaces (Zhang et al., 2020), wetlands (Zhang et al., 2023b), forests (Tang et al., 2023), or croplands (Blickensdörfer et al., 2022; Sun et al., 2024).

The free access to Sentinel imagery and to the powerful cloud computation platform provide ample opportunity for global land-cover mapping at 10 m. In this work, we developed an accurate and novel global 10-m land-cover product (GLC_FCS10) containing 30 fine land-cover types from Sentinel-1 and Sentinel-2 time-series imagery. To achieve this goal, we propose: 1) a hierarchical land-cover mapping framework to decrease the uncertainties of impervious surfaces and wetlands; 2) to combine the prior multisource land-cover datasets and the metric centroid to automatically generate a globally distributed and high-confidence 10-m training pool; 3) to integrate time-series Sentinel-2 optical and Sentinel-1 SAR data for producing the new GLC_FCS10 on the GEE platform; 4) to comprehensively compare the developed GLC_FCS10 with several previous products.

## 2. Datasets

### 2.1 Satellite imagery

All available Sentinel-2 surface reflectance imagery in 2023 were atmospherically corrected using the Sen2Cor tool, and the corrected images were then stored on the GEE (google earth engine) platform. These imagery contain 12 spectral bands from visible to shortwave infrared and have a revisiting period of 5 days (Berger et al., 2012; Radeloff et al., 2024). In this work, the four 10-m visible and near-infrared bands and six 20-m red edges and shortwave infrared bands were used, while the two 60-m bands of aerosols and water vapor were excluded to minimize atmospheric effects. The six 20-m reflectance imagery bands were resampled to 10-m using the bilinear resampling method (Berger et al., 2012). Any poor-quality pixels, including clouds and shadows, were masked using the quality control band (QA60) and the cloud probability product.

Sentinel-1 has a dual-polarization (VV and VH) C-band SAR instrument with a revisiting period of 6 days after launching of Sentinel-1B (Torres et al., 2012). In this work, all Sentinel-1 imagery in 2023 were obtained through the GEE platform, which have been preprocessed through radiometric calibration, thermal noise removal, and terrain correction, and further resampled their resolution of 5 m × 20 m into 10 m × 10 m using Sentinel-1 Toolbox when archived on the GEE platform.

Some previous works have demonstrated that topographical data can provide auxiliary and useful information in land-cover mapping (Zhang et al., 2023b), currently, global 10 m digital elevation model (DEM) is not yet available. In this work, we used the 30-m ASTER GDEM, which has low vertical error of 0.7 m (Tachikawa et al., 2011), to obtained the elevation, slope, and aspect after bilinear resampling to 10 m × 10 m.

### 2.2 Prior land-cover products

#### 2.2.1 Impervious surface products

Impervious surfaces are characterized by sparse spatial distribution and complicated spectral and spatial heterogeneities; thus, it should be treated independently. Its training samples are generated from five previous products: 1) The Global 30-m Impervious Surface Dynamic (GISD30) dataset, developed with the combination of

spectral generalization and sample migration during 1985-2020 with the interval of 5-years, achieves a fulfilling accuracy of 90.1% (Zhang et al., 2022). 2) The Global Impervious Surface Area (GISA 2.0) dataset, produced by considering the inconsistency among four existing products, is an annual time-series impervious surface maps during 1985-2018 with the F1-score of 0.935 (Huang et al., 2022). 3) The 10 m impervious surface layer in ESA WorldCover dataset was generated by the supervision classification from time-series Sentinel-1 and Sentinel-2 imagery (Zanaga et al., 2021). It was demonstrated to achieve the great performance on impervious surfaces with producer's accuracy of 82.99% (Zhao et al., 2023). 4) The impervious surface layer in ESRI Land Cover is developed from time-series Sentinel-2 imagery and the deep-learning (Karra et al., 2021), and achieve high producer's accuracy of 88.42% (Zhao et al., 2023) .5) Global urban boundary dataset (GUB) is generated by the combination of cellular-automata and morphological approach, and shows a good agreement with the results from human interpretation (Li et al., 2020).

**Table 1**. The characteristics of prior land-cover products.

| Category | Dataset name | Resolution | Year | Coverage | Reference |
|---|---|---|---|---|---|
| Impervious surface | GISD30 | 30 | 1985–2020 | Global | Zhang et al. (2022) |
| | GISA 2.0 | 30 | 1985–2019 | Global | Huang et al. (2022) |
| | Imp-ESA_LC | 10 | 2021 | Global | Zanaga et al. (2021) |
| | Imp-ESRI_LC | 10 | 2023 | Global | Karra et al. (2021) |
| | GUB | - | 2020 | Global | Li et al. (2020) |
| Wetland | GWL_FCS30D | 30 | 2000–2022 | Global | Zhang et al. (2024b) |
| | National Wetland Inventory | 30 | 2019 | United States | Wilen and Bates (1995) |
| | Global Mangrove Watch | 30 | 1996–2020 | Global | Bunting et al. (2022) |
| | Global tidal flat products | 30 | 2000–2022 | Global | Zhang et al. (2023a) |
| | Global tidal marsh dataset | 10 | 2020 | Global | Worthington et al. (2024) |
| Land-cover | GLC_FCS30D | 30 | 1985–2022 | Global | Zhang et al. (2024c) |
| | Global oil palm dataset | 30 | 1990–2021 | Global | Descals et al. (2024) |
| | Global plantation map | 30 | 1982–2020 | Global | Du et al. (2022) |

**2.2.2 Wetland products**

Because almost all global land-cover products have large uncertainties in wetlands identification (Zhang et al., 2023b), wetlands should also be treated as independent land-cover type. The wetland training samples were also derived from three existing wetland thematic products, including: 1) the GWL_FCS30D is an annual global wetland products containing 8 wetland subcategories (5 inland and 3 coastal wetland subcategories), and achieves an overall accuracy of 86.95 ± 0.44% (Zhang et al., 2024b). 2) The NWI (National Wetland Inventory) is national wetland thematic products covering the whole United States and containing 8 wetland subcategories (Wilen and Bates, 1995). 3) The GMW (Global Mangrove Watch) and GSM10 (global tidal marsh dataset) provide the spatial patterns of mangrove and salt marsh with the overall accuracy of 87.4% and 85% (Bunting et al., 2022; Worthington et al., 2024), and the GTF30 (Global tidal flat products), generated by the combination of a new low-tide spectral index and multisource classification method, achieved the overall accuracy of 90.34% and covered the period of 2000-2022 (Zhang et al., 2023a).

**2.2.3 GLC_FCS30D land-cover dynamic products**

Except for wetlands and impervious surfaces, training samples for the remaining non-wetland natural land-cover types are generated from the GLC_FCS30D dataset. It was developed from the combination of a continuous change detection algorithm with an adaptive updating strategy and had 80.88% (±0.27%) accuracy covering the period of 1985–2022 with 35 fine land-cover subcategories. The dataset has high temporal stability in the European Union and

United States (Zhang et al., 2024c). In this work, we leverage this dynamic product to generate confident training
samples for non-wetland natural land covers, as described in the Section 3.1.

**2.2.4 Tree-cover cropland datasets**

It is noteworthy that the tree-cover cropland was only mapped in certain regions rather than globally in the
GLC_FCS30D (Zhang et al., 2021; Zhang et al., 2024c); thus, global oil palm and plantation datasets are also used
to identify tree-cover cropland. The global oil palm dataset is a time-series covering 1990–2020 and exceeds 91%
accuracy for industrial plantations and 71% accuracy for smallholders (Descals et al., 2024). A global plantation map
was generated by combining some prior global plantation products and the LandTrendr method, and has an F1 score
of 86.83% with ±5 years tolerance (Du et al., 2022).

## 3. Methodology

To achieve high quality with detailed categorizations in global 10-m land-cover mapping, a hierarchical land-
cover mapping methodology has been proposed. It leverages multisource prior land-cover products and time-series
satellite observations, and gives more attention to impervious surfaces and wetlands by importing more prior
knowledge and adding sufficient high-confidence training samples. Notably, the reasons why we separated
impervious surfaces and wetlands from other land cover types include: 1) impervious surfaces are structurally
different from natural land covers (Huang et al., 2022; Zhang et al., 2022); 2) wetlands are a highly zonal land-cover
type (concentrating in low-lying areas) with extremely complex spectra and heterogeneities due to changes in
phenology and water-levels (Mao et al., 2020; Zhang et al., 2023b); 3) some previous studies emphasized that taking
additional measures (such as: importing more prior knowledge, thematic mapping strategy and so on) was an effective
means of improving the accuracy of wetlands and impervious surfaces (Gong et al., 2020; Zhang et al., 2023b).
Figure 1 presents a flowchart of the proposed method, which involves four procedures: generating hierarchical and
globally distributed training samples from prior products, compositing multisource and multitemporal training
features from time-series Sentinel 1&2 imagery, hierarchical land-cover mapping using local adaptive classifications,
and accuracy assessment and cross-comparisons.

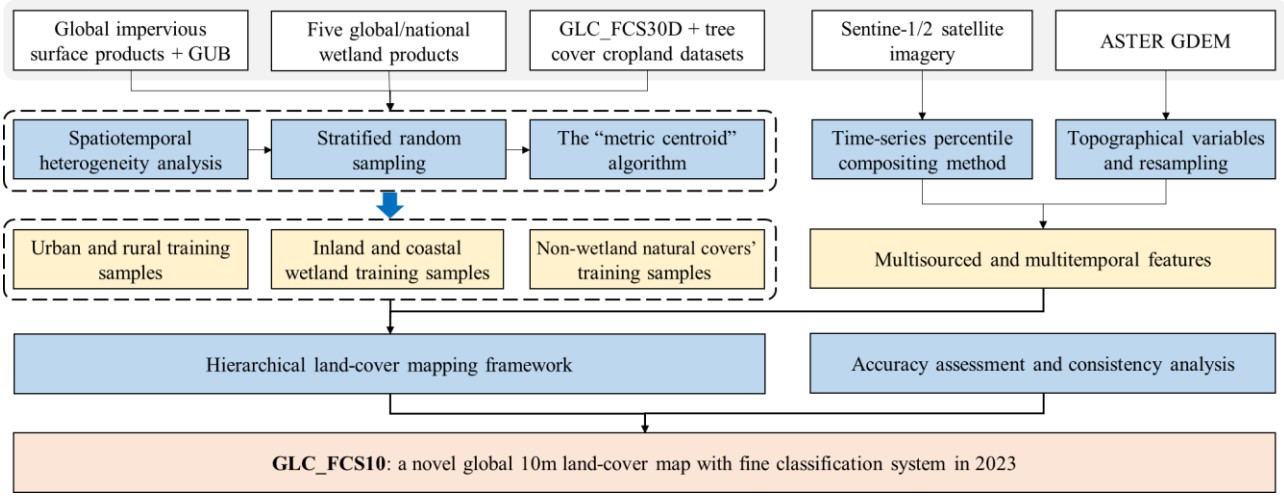

**Figure 1**. The flowchart of the proposed method for hierarchical land-cover mapping.

## 3.1 The description of the classification system

In this work, we develop a novel global 10-m land-cover dataset with a fine classification system (FCS). Table 2
presents the main characteristics of this FCS and its correspondence with the basic classification system. It contains
30 fine land-cover types and emphasizes the forest- and wetland-related subcategories, which are further subdivided
into 10 and 7 subcategories, respectively. The diversity of this fine classification system results from importing the
GWL_FCS30D (Zhang et al., 2024b) and GLC_FCS30D (Zhang et al., 2024c) products. It should be noted that the
detailed descriptions about each fine land-cover type have been supplemented at the Table S1.
**Table 2**. The characteristics of the fine classification system in the GLC_FCS10.

| Basic classification system | Detailed validation system | Fine classification system |
|---|---|---|
| Cropland | Rainfed cropland | Herbaceous rainfed cropland |
| | | Tree or shrub covered rainfed cropland (orchard, oil palm…) |
| | Irrigated cropland | Irrigated cropland |
| Forest | Evergreen broadleaved forest | Closed evergreen broadleaved forest |
| | | Open evergreen broadleaved forest |
| | Deciduous broadleaved forest | Closed deciduous broadleaved forest |
| | | Open deciduous broadleaved forest |
| | Evergreen needleleaved forest | Closed evergreen needleleaved forest |
| | | Open evergreen needleleaved forest |
| | Deciduous needleleaved forest | Closed deciduous needleleaved forest |
| | | Open deciduous needleleaved forest |
| | Mixed-leaf forest | Closed mixed-leaf forest |
| | | Open mixed-leaf forest |
| Shrubland | Shrubland | Evergreen shrubland |
| | | Deciduous shrubland |
| Grassland | Grassland | Grassland |
| Tundra | Tundra | Lichens and mosses |
| Wetland | Wetland | Swamp |
| | | Marsh |
| | | Lake/river flat |
| | | Saline |
| | | Mangrove forest |
| | | Salt marsh |
| | | Tidal flat |
| Impervious surfaces | Impervious surfaces | Urban impervious surfaces |
| | | Rural impervious surfaces |
| Bare areas | Sparse vegetation | Sparse vegetation |
| | Bare areas | Bare areas |
| Water | Water | Water |
| Permanent ice and snow | Permanent ice and snow | Permanent ice and snow |

## 3.2 Generating hierarchical training samples

To ensure quality in global 10-m land-cover mapping, land surfaces are hierarchically divided into impervious
surface, wetland, and non-wetland natural land-cover types. Their corresponding training samples also need to be
generated independently. Because training sample quality greatly affects land-cover mapping performance (Foody
and Arora, 2010; Zhang and Roy, 2017), generating confident and globally distributed training samples is a
prerequisite for generating the GLC_FCS10.

### 3.2.1 Training areas of impervious surfaces

Regarding the training samples of impervious surfaces, we combine four prior global 10-m or 30-m impervious
surface products (GISA 2.0, GISD30, Imp-ESA_LC, and Imp-ESRI_LC) and one urban boundary dataset (GUB) to
automatically generate the training samples. Specifically, because the previous studies have demonstrated high-
accuracies of three impervious surface products (Huang et al., 2022; Zanaga et al., 2021; Zhang et al., 2022)and high
producer's accuracy of Imp-ESRI_LC (Zhao et al., 2023), the areas marked as impervious surfaces by all four
products (GISD30-2020, GISA2.0-2019, Imp-ESA_LC-2021, and Imp-ESRI_LC-2023) are selected as candidate
areas for generating the training samples (*TrainCanArea_imp*) in Eq. (1).

$$TrainCanArea_{imp} = GISD30 \cap GISA2.0 \cap (\text{Imp-ESA}_{LC}) \cap (\text{Imp-ESRI\_LC}) \qquad (1)$$

Afterward, we further consider the uneven distribution of rural and urban impervious surfaces as well as their spectral
variability. If random sampling is used to obtain training samples from the *TrainCanArea_imp*, rural impervious
surfaces are underrepresented due to their sparse distribution, thus, the GUB urban boundary dataset for 2020 is
further used to divide the *TrainCanArea_imp* into urban (*TrainCanArea_urban*) and rural areas
(*TrainCanArea_rural*).
Beyond the confident impervious surface areas, it is equally important to identify high-quality natural land-cover
types (Zhang et al., 2024a). To avoid confusion between natural land-cover types and impervious surfaces, the
maximum impervious surface boundary (*MaxBound_imp*) is also generated. The training samples for natural land-
cover types should be located outside of the *MaxBound_imp*, i.e., some inner-city areas, easily misclassified or
confused with impervious surfaces, will be excluded because Imp-ESRI_LC exhibits extensive patches of the
impervious surface and lacks spatial details (Wang et al., 2024; Xu et al., 2024b). To determine *MaxBound_imp*, the
union of the four products is applied as:

$$MaxBound_{imp} = GISD30 \cup GISA2.0 \cup (\text{Imp-ESA}_{LC}) \cup (\text{Imp-ESRI\_LC}) \qquad (2)$$

**3.2.2 Training areas of wetland**
In this work, wetlands are divided into four inland subcategories (swamp, marsh, river/lake flats, and saline) and
three coastal wetland subcategories (mangrove, salt marsh, and tidal flats in **Table 2**). Because coastal wetlands have
a more pronounced zonation, and the global coastal wetland mapping have make great progresses while the works of
global inland wetland mapping is still sparse (Wang et al., 2023), thus, the generation of wetland training candidate
areas further distinguishes between inland and coastal wetlands.
The time-series GWL_FCS30D wetland product covering the period of 2000–2022 is used to derive the inland
wetland training candidate areas (Zhang et al., 2024b). Because temporally stable areas achieve higher accuracy
(Yang and Huang, 2021), a temporally stable analysis is applied to the GWL_FCS30D, and only those stable areas
where wetland subcategories do not change during 2000–2022 are retained (yielding the training area,
*TrainCanArea_Inwet*). Then, because adjacent land-cover areas are easier to suffer from the higher misclassifications
(Radoux et al., 2014) and to the impact of the satellite geolocation error (Zhang and Roy, 2017), a spatial filter with
a local window of 3 pixels × 3 pixels is applied to *TrainCanArea_Inwet* to retain spatially homogeneous areas as the
training areas. Further, the integration of multiple wetland products can further improve sample quality, but there are
few high-quality wetland products that have been publicly shared. In this study, only the National Wetland Inventory
for 2019 is imported to optimize the swamp and marsh land-cover types in *TrainCanArea_Inwet* over the United
States because the National Wetland Inventory does not identify river/lake flats or saline subcategories. Namely, the
areas identified as swamp and marsh in *TrainCanArea_Inwet* and the National Wetland Inventory are retained. .
As for the three coastal wetland subcategories, their training areas are generated from the combination of
GWL_FCS30D and three coastal wetland products (GMW, GTF30, and GSM10 in **Table 1**). We identify temporally
stable coastal wetland areas from GWL_FCS30D, GMW, and GTF30 through time-series analysis and label them
GWL_stable, GMW_stable, and GTF_stable. Then, we intersect GWL_stable with GMW_stable to generate
mangrove forest training areas and intersect GWL_stable with GTF_stable to generate tidal flat training areas. Next,
as the GSM10 only provides salt marsh maps in 2020, the salt marsh training areas are selected as the intersection
between GWL_stable and GSM10. The mangrove forest, tidal flat, and salt marsh training areas are grouped as
*TrainCanArea_Cowet*.
Last, the maximum wetland boundary (*MaxBound_wet*) is also necessary for the subsequent identification of
training areas for non-wetland natural land-cover types. *MaxBound_wet* is determined as the union of the five global
wetland products:

$$MaxBound_{wet} = GWL\_FCS30D \cup NWI \cup GMW \cup GTF30 \cup GSM10 \qquad (3)$$

### 3.2.3 Training areas of non-wetland natural land-covers

Many previous works have emphasized that these spatiotemporally stable areas always performed higher
mapping accuracy (Zhang and Roy, 2017; Zhang et al., 2024c). In this work, the time-series global 30-m land-cover
dynamic product (GLC_FCS30D), covering the period of 1985–2022 is used. Specifically, three measures are taken
to identify spatiotemporally stable areas of non-wetland natural land-cover types from GLC_FCS30D: 1) a time-
series consistency analysis is applied to the GLC_FCS30D, and only stable areas during 1985–2022 will be retained
as *TrainCanArea_NLCs*. 2) The $MaxBound_{imp}$ and $MaxBound_{wet}$ are imported to mask the *TrainCanArea_NLCs*,
i.e., the training areas for non-wetland natural land-cover types should be located outside of $MaxBound_{imp}$ and
$MaxBound_{wet}$. The aim of this step is to minimize confusion between non-wetland natural land-cover types and
these two land-cover types. 3) A morphological erosion filter with a local window of 3 pixels × 3 pixels is used to
find the spatially homogeneous areas for non-wetland natural land-cover types. In addition, it should be noted that
the TrainCanArea_NLCs represents the stable areas during 1985-2022, and there is still one-year interval with the
land-cover mapping year in 2023. Fortunately, the ongoing updating of GLC_FCS30D is still in progress, the land-
cover change masks during 2022-2023 have been finished, and which are used to guarantee the temporal consistency
between prior land-cover products with the training areas.
As mentioned in Section 2.2.4, the training areas for tree-cover cropland (oil palm, orchards, etc.) from the
GLC_FCS30D do not cover the globe. These cropland training areas are therefore divided into herbaceous rainfed
cropland and tree- or shrub-cover cropland. Because the global oil palm and global plantation datasets provide the
plantation years of oil palm and orchards at 30 m, we overlap the training areas of rainfed cropland, oil palm, and
orchard plantation from the global plantation dataset to extract the training areas for tree-cover cropland. Then, to
minimize error, the tree-cover cropland training areas are further filtered using a local window of 3 pixels × 3 pixels
to ensure spatial homogeneity of tree-cover cropland training areas.

### 3.2.4 Generating a globally distributed training pool from stratified sampling

Although we take a series of measures to ensure training area quality (including: *TrainCanArea_urban,*
*TrainCanArea_rural, TrainCanArea_Inwet, TrainCanArea_Cowet,* and *TrainCanArea_NLCs*), how to generate
training samples from the training areas needs to address the following two aspects.
First, the distribution and balance of training samples greatly affect the subsequent land-cover classification
(Ghorbanian et al., 2020; Jin et al., 2014; Mellor et al., 2015; Pelletier et al., 2017; Zhu et al., 2016). There are two
options to allocate the sample distribution: equal or area-fraction allocation (Zhang et al., 2021). Equal distribution
means that all land-cover types have the same number of training samples, i.e., the sample sizes of sparse land covers
will be augmented while those of the abundant land covers will be suppressed. In contrast, area-fraction distribution
allocates the sample size according to the land-cover area of each type, that is, abundant land covers have larger
sample sizes while sparse land-cover types have smaller sample sizes (Zhu et al., 2016). Because impervious surfaces
and wetlands are sparser than natural land-cover types and are independently generated, equal-distribution allocation
is suitable to enhance the training samples' ability to characterize these two land-cover types. As for the non-wetland
natural land-covers, the area-fraction allocation is more appropriate for the non-wetland natural land-cover types
because we want to optimize results for all non-wetland natural land-cover types rather than a single land-cover type.
Meanwhile, to avoid sample size imbalance in the area-fraction allocation, maximum and minimum sample sizes of
8000 pixels and 600 pixels are chosen for the abundant and sparse land-cover types, respectively (suggested by the
work of Zhu et al. (2016)).
Second, most high-quality training samples (except for those for impervious surfaces) are derived from the 30-
m training areas, so there is also a need to reduce the 30-m training samples to 10-m samples to achieve a global 10-
m land-cover map. In this work, the "metric centroid" method is adopted, which had been used to downscale 500-m
training samples from MCD12Q1 to 30-m in the work of Zhang and Roy (2017). Specifically, as each 30-m pixel
corresponds to 3 × 3 10-m pixels, we first find the centroid from these nine pixels as $P_{centroid}$ through spectral
averaging, and then the point with the smallest absolute distance with $P_{centroid}$ was chosen as the optimal downscaled
10-m sample point [**Eq. (4)**].

$$P_i = \underset{i}{\mathrm{argmin}}\left(\left|\boldsymbol{\rho}_{P_i} - \boldsymbol{\rho}_{P_{centroid}}\right|\right), \boldsymbol{\rho}_{P_{centroid}} = \frac{1}{9}\sum_{j=1}^{9}\frac{\boldsymbol{\rho}_{P_j}}{9} \tag{4}$$

Where $\boldsymbol{\rho}_{P_i}$ is the spectra value of composited Sentinel-2 training features (See Section 3.3) at pixel $P_i$. If more than
one point in the nine pixels has the same minimum absolute distance, then we pick randomly from among them.

### 3.3 Compositing multisourced training features

285        In addition to high-confidence training samples, how to generate these multisource training features is also
important (Dong et al., 2015; Jin et al., 2023; Yang and Huang, 2021). In this work, we composite multitemporal
optical and SAR information from Sentinel-1 and Sentinel-2 time-series observations. First, because of the
overlapping orbits of the satellites and the effects of clouds and shadows, there are substantial differences in the clear-
sky observations in different regions. Compositing methods help to obtain dimensionally consistent spectral-
phenological features. The percentile-based statistical multitemporal compositing method attracts attention because
of its robustness, efficiency, and simplicity (Azzari and Lobell, 2017). The basic principle of this method is to
rearrange intra-annual time-series reflectance according to mathematical magnitude and take the corresponding
quartiles to reflect the phenological variation of the time-series and suppress the noise interference such as clouds
and shadows (Hansen et al., 2014). Many previous studies have demonstrated its ability to flexibly balance noise
removal and signal retention, that is, it can efficiently capture the phenological variations with less prior knowledge
and is also robust to the residual cloud and shadows (Hansen et al., 2014; Zhang and Roy, 2017). Thus, in this study,
time-series Sentinel-2 images are composited into the 10th, 30th, 50th, 70th, and 90th percentiles for their 10 optical
bands from visible to shortwave infrared and three typical indexes [NDVI, NDWI, and LSWI in **Eq. (5)**] using the
percentile-based compositing method. The 10th and 90th percentiles are selected to represent the poles of time-series
variations and also suppress the effects of residual cloud and shadow, and the other three percentiles can partly reflect
the phenological variations (Xie et al., 2020). Meanwhile, another major advantage of 10-m Sentinel-2 imagery is
that it provides clearer textural features, so we generate time-series texture features from the five percentiles in the
NIR bands using the gray level co-occurrence matrix. Only the texture features in the NIR band are extracted to avoid
redundancy, because of the texture similarity within different spectral bands (Rodriguez-Galiano et al., 2012).

$$NDVI = \frac{\rho_{NIR} - \rho_r}{\rho_{NIR} + \rho_r}, LSWI = \frac{\rho_{NIR} - \rho_{SWIR1}}{\rho_{NIR} + \rho_{SWIR1}} \ and \ NBWI = \frac{\rho_{green} - \rho_{SWIR1}}{\rho_{green} + \rho_{SWIR1}} \tag{5}$$

where $\rho_{green}, \rho_r, \rho_{NIR}, \rho_{SWIR1}$ are the spectral bands of green, red, NIR, and SWIR1 in the Sentinel-2 imagery.
306        Then, because SAR signals are sensitive to changes in surface water dynamics and spatial structure, it is also
necessary to extract multitemporal SAR features from Sentinel-1 imagery (Bullock et al., 2022; Dabrowska-Zielinska
et al., 2018; Zhang et al., 2020). The percentile-based composited method is also used to capture the time-series
variabilities of VV and VH at the 10th, 30th, 50th, 70th, and 90th percentiles. In summary, a total of 10 SAR features
are composited from time-series Sentinel-1 observations.
311        Afterward, because some land-cover types are characterized by important topographic distribution (e.g.,
permanent snow and ice are clustered in high elevation areas, croplands and impervious surfaces usually locate on
these flat areas), the topographical variables (slope, aspect and elevation), generated from the resampled ASTER
GDEM dataset, are collected into the multisourced training features. It should be noted that some 5 × 5 geographical
tiles do not have sufficient Sentinel-1 observational data in 2023 since Sentinel-1B is retired in 2022, the
corresponding tiles use only Sentinel-2 and topographic data.

## 3.4 Hierarchical land-cover mapping

A major advantage of hierarchical land-cover mapping is the ability to improve the characterization of complex
land-covers with independent models, the major flowchart is illustrated at the Figure 2. In this work, we first separate
global land-cover types into impervious surfaces and natural land-cover types, then identify the wetlands from among
the natural land-covers, and finally classify the remaining non-wetland natural land-cover types into 20 land-cover
subcategories.

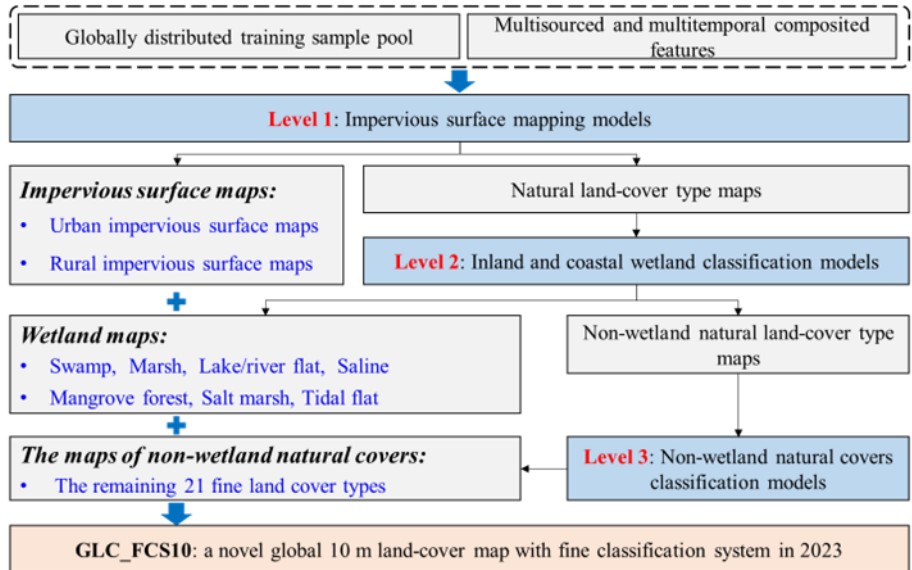

**Figure 2.** The detailed flowchart of hierarchical land-cover mapping algorithm by integrating globally distributed
training samples and multisourced composited features.

### 3.4.1 The separation of impervious surfaces and natural surfaces

To separate impervious surfaces and natural surfaces, we rely on the globally distributed training samples
(Section 3.2.4) and the combination of multitemporal optical and SAR features. Specifically, because we divide
impervious surfaces training samples into rural and urban samples and design the equal distribution to enhance the
training samples' ability to characterize impervious surfaces. The ratio of urban samples, rural samples, and natural
surfaces is 1:1:1 for each $5 \times 5$ geographical tile. Meanwhile, in terms of the sample size of each class, some previous
studies have quantified the relationship between sample size and mapping accuracy (Foody, 2009; Li et al., 2014),
and suggested a minimum size of 600 and maximum size of 8000 for these sparse and abundant land-cover types
(Zhu et al., 2016). In this study, after considering the trade-off between sample representativeness with mapping
efficiency, the sample size of each class was selected as 5000, which was also consistent with the work of Zhang et
al. (2022) in monitoring the impervious surface dynamics.
Then, we split the globe into 984 $5 \times 5$ geographical tiles (approximately 556 km × 556 km on the equator,
illustrating on the Figure S1), because some studies emphasized that the local adaptive modeling usually achieves
better mapping accuracy than single land-cover global modeling (Zhang et al., 2021), and previous works of Zhang
and Roy (2017) and Zhang et al. (2019) have explained that the training samples of sparse land-cover types in a small
geographical grid were usually missed or greatly sparse. Thus, after balancing the training sample volume, mapping
accuracy, and the limitation of GEE platform, the local modeling tile size of $5 \times 5$, similar to the works of (Zhang
et al., 2021; Zhang et al., 2024c), were used. When building the training model for each $5 \times 5$ geographical tile, we
also import training samples within their spatial neighborhood of 3 × 3 tiles to ensure spatial consistency over the
adjacent tiles. Since the $MaxBound_{imp}$ (Eq. (2)) provides the maximum potential areas of impervious surfaces

because of the overestimation problem of Imp-ESRI_LC (Wang et al., 2024; Xu et al., 2024b), all identified impervious surfaces should be within the $MaxBound_{imp}$. Afterward, we can produce 984 5 × 5 impervious surface and natural land cover maps using the local adaptive modeling strategy. In addition, although we divide the training samples into urban and rural samples, there is serious confusion between urban and rural areas in the classification maps because they share similar spectral and SAR characteristics. We therefore consider the two subcategories to be inseparable at the classification stage. Correspondingly, inspired by the work of Li et al. (2020) who used the cellular automata and morphological approaches to accurately capture urban boundaries, this method is also applied in this work to distinguish urban and rural impervious surfaces.

Notably, in terms of the selection of classifier at the local adaptive modeling, the random forest classification model (including the later section 3.4.2 and 3.4.3) is used. The random forest has some advantages over other traditional classifiers, such as managing high-dimensional data more efficiently, having higher mapping robustness, being insensitive to parameter settings, and avoiding overfitting problems effectively (Belgiu and Drăguţ, 2016; Breiman, 2001). In terms of its parameter settings, the random forest only has two adjustable parameters, and the variations of these two parameters have little effect on the performance of the random forest model (Du et al., 2015; Gislason et al., 2006), thus, the default parameter settings are applied to train the random forest models on the GEE platform.

### 3.4.2 The separation of wetland and non-wetland natural land-cover types

In terms of how to identify the fine wetland subcategories from natural land-covers, we use the stratified wetland mapping algorithm to independently distinguish coastal wetlands and inland wetlands. Wetlands are divided into four inland and three coastal wetland subcategories (in **Table 2**), and equal-distribution sampling is used to enhance the training samples' ability to characterize wetlands. Additionally, since some non-wetland land-cover types also reflected the similar spectral characteristics with the wetlands, for example, the swamp and the forest/shrubland shared similar vegetation spectra during the peak growth period, while the marsh and cropland/grassland exhibited the characteristics of herbaceous vegetations, and the river flats also performed the spectral characteristics of bare land during the dry seasons (Zhang et al., 2023b). Thus, the approximate ratio of inland wetlands, coastal wetlands, and non-wetlands (including water body, forest/shrubland, cropland/grassland, bare land, and others) is 4:3:5 in areas where they coexist. Then, because coastal wetlands have a more pronounced zonation, we can obtain their maximum coverage through the union of some previous coastal wetland products, as Eq. (6).

$$MaxBound_{Cos\_wet} = GWL\_FCS30D\_Coastal \cup GMW \cup GTF30 \cup GSM10 \qquad (6)$$

When building the wetland random forest classification models for each 5 × 5 geographical tile, we first train the coastal wetland classification model using the coastal wetland and non-wetland training samples within their spatial neighborhood of 3 × 3 tiles, and combine multisourced training features to identify the spatial distribution of coastal wetlands within the $MaxBound_{Cwet}$; i.e., all coastal wetland pixels should be within the $MaxBound_{Cwet}$, otherwise, they would be corrected.

Afterwards, the inland wetland and non-wetland training samples are used to train another random forest classification model, and the remaining areas are further classified as four inland wetland subcategories and non-wetland natural land-cover types using the inland wetland classification model.

### 3.4.3 Mapping of non-wetland natural land-cover types

After classifying the impervious surface and wetlands using the hierarchical land-cover mapping, we now need to classify the remaining non-wetland natural land-cover types. Like the previous mapping processes, the local adaptive random forest models are trained for each 5 × 5 geographical tile using the corresponding training samples within the spatial neighborhood of 3 × 3 tiles. The non-wetland natural land-cover types are classified through a combination of trained random forest models and multisource training features. Lastly, after overlapping the

hierarchical maps for impervious surface, wetland, and non-wetland natural land-cover types, we can obtain 10-m
land-cover maps with a fine classification system.

## 3.5 Accuracy assessment and cross-comparison

To comprehensively assess the performance of our developed GLC_FCS10 products, a globally distributed
validation dataset and one third-party validation dataset are collected to quantify the accuracy metrics. First, the
global validation dataset, guided by the work of (Zhao et al. (2023), is collected through stratified random sampling
and visual interpretation from high-resolution remote sensing imagery in 2023. Figure 3 illustrates the spatial
distribution of the global validation dataset; it contains 56121 globally distributed validation points and covers 16
land-cover types. Next, the Land Cover Monitoring, Assessment, and Projection Collection (LCMAP) validation
dataset (called LCMAP_Val), as a national third-party validation dataset, contains 16082 nationally distributed
validation points during 1985–2021 (Stehman et al., 2021). In this work, the LCMAP_Val in 2021 is also updated to
2023 through visual interpretation. Afterward, the confusion matrix and four accuracy metrics are calculated,
including: the overall accuracy (O.A.) and kappa coefficient (measuring the comprehensive performance) and the
producer accuracy (P.A.), and the user accuracy (U.A.), which measure the commission and omission errors for each
land-cover type to quantify the accuracy of GLC_FCS10 (Foody and Arora, 2010; Liu et al., 2007; Nelson et al.,
403 2021).

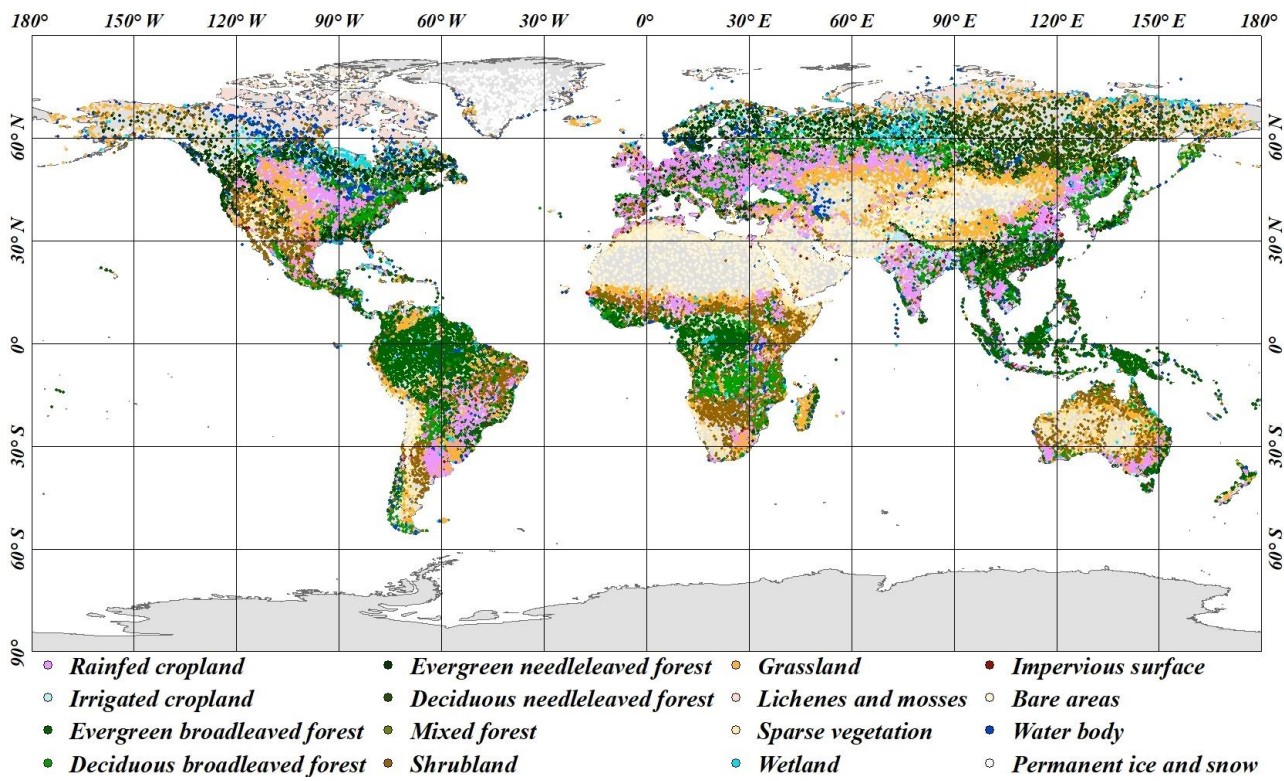


**Figure 3.** The spatial distribution of global validation samples containing 16 land-cover types in 2023.
In addition, to qualitatively investigate the performance of GLC_FCS10, three global 10-m land-cover products
[ESRI_LC (Karra et al., 2021), ESA WorldCover (Zanaga et al., 2021), and FROM_GLC10 (Gong et al., 2019)], and
one 30-m land-cover product [GLC_FCS30 (Zhang et al., 2021)] are collected as comparative products. None of
these five data products have been updated to 2023. Their latest available data will be collected for our comparative
analysis.

## 4. Results and discussions

## 4.1 Overview of the GLC_FCS10 map

Figure 4 illustrates the spatial distribution of the GLC_FCS10 land-cover map with 30 fine land-cover types in 2023. Overall, it accurately charactersizes global land-cover patterns, i.e., forests concentrate in tropical rainforest regions and cold temperate forest zones in the northern hemisphere, cropland is found in low-lying plains areas such as the North China Plain, Central Plains of the United States, Central Eurasia, and bare land and grassland are distributed in arid and semiarid areas. Meanwhile, because a characteristic of the GLC_FCS10 is its diverse classification system, we can see that broadleaved forests are found in low- and medium-altitude regions, while needle-leaved forests are distributed in cold temperate zones as well as in high-altitude areas.

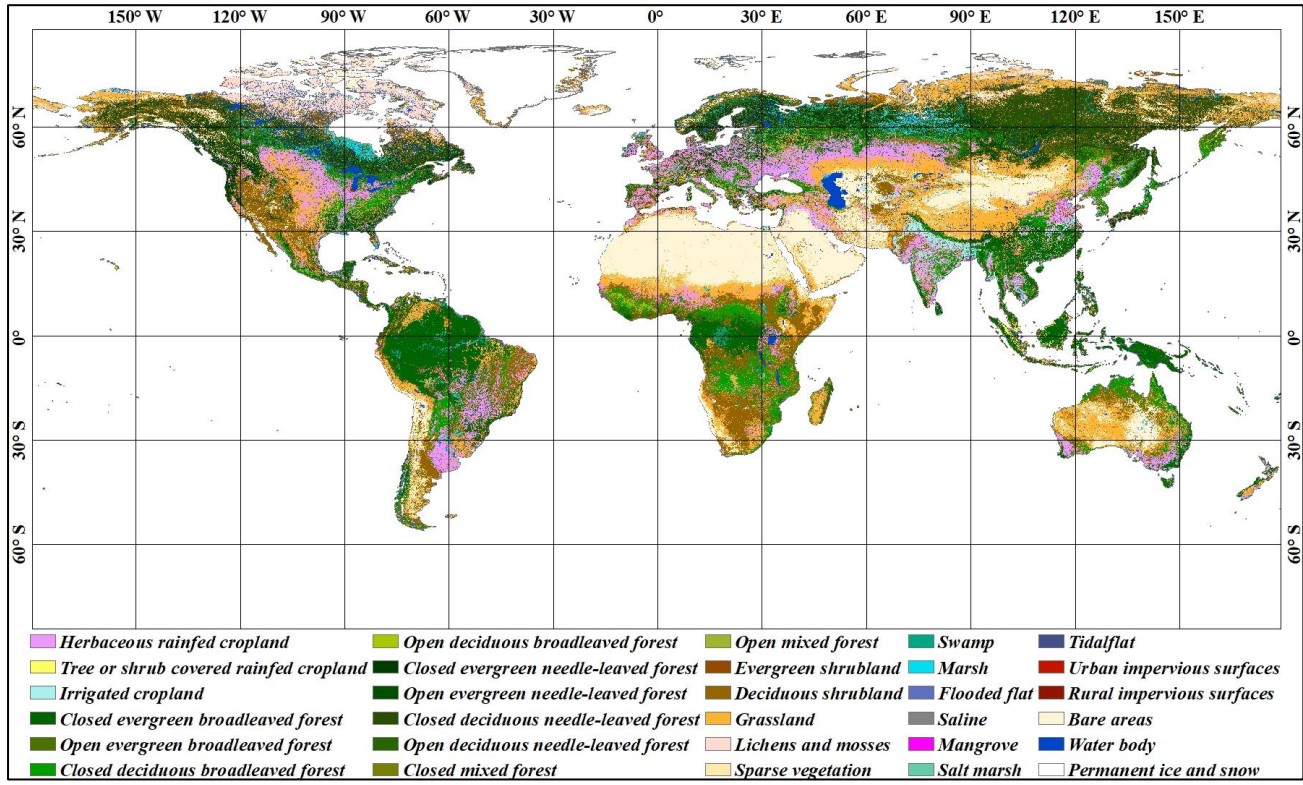

**Figure 4**. Spatial distribution of the 30 fine land-cover types in GLC_FCS10 land-cover map in 2023.

## 4.2 Accuracy assessment

### 4.2.1 Global-scale accuracy assessment

Table 3 presents the confusion matrix between GLC_FCS10 and the 56121 globally distributed validation points for 10 major land-cover types (corresponding to the basic classification system in Table 2). Overall, GLC_FCS10 achieves an O.A. of 83.16% and a kappa coefficient of 0.789. For specific land-cover types, permanent snow and ice, water bodies, forest, impervious surfaces, and cropland perform the best, with the corresponding U.A. and P.A. values approximating or exceeding 90%. Their high accuracies stem from the distinct spectral properties inherent to water bodies and permanent ice and snow, the abundant coverage of cropland and forest, and hierarchical land-cover mapping for impervious surfaces. However, shrubland, grassland, tundra, and wetlands suffer obvious misclassifications, in which the shrubland has the lowest U.A. of 67.04% and wetland has the lowest P.A. of 53.69%. There are considerable confusions between shrubland, grassland, and bare areas because they share similar spectral characteristics and coexist in arid and semiarid areas. Wetlands have the lowest P.A. due to the confusions between wetlands, water bodies, forest, and grassland. Wetlands have complicated and heterogeneous spectral and temporal variations, thus, the swamp subcategory is easily confused with forest, and the marsh subcategory shares spectral characteristics with grasslands (Zhang et al. (2023b).

**Table 3**. The confusion matrix between GLC_FCS10 and the globally distributed validation dataset for 10 major
land-cover types.

|  | Crop | Forest | Shrub | Grass | Tundra | Wetland | Impervious | Barren | Water | Ice-Snow | Total | U.A. |
|---|---|---|---|---|---|---|---|---|---|---|---|---|
| **Crop** | 8442 | 339 | 588 | 768 | 0 | 53 | 39 | 46 | 9 | 0 | 10284 | 82.09 |
| **Forest** | 161 | 18342 | 1191 | 189 | 0 | 250 | 2 | 12 | 4 | 0 | 20151 | 91.02 |
| **Shrub** | 190 | 701 | 4091 | 922 | 12 | 72 | 4 | 109 | 1 | 0 | 6102 | 67.04 |
| **Grass** | 673 | 255 | 275 | 5817 | 33 | 170 | 7 | 391 | 4 | 1 | 7626 | 76.28 |
| **Tundra** | 0 | 25 | 78 | 153 | 805 | 2 | 0 | 61 | 0 | 0 | 1124 | 71.62 |
| **Wetland** | 10 | 74 | 136 | 100 | 14 | 946 | 1 | 23 | 30 | 0 | 1334 | 70.91 |
| **Impervious** | 20 | 6 | 11 | 28 | 0 | 1 | 902 | 5 | 0 | 0 | 973 | 92.70 |
| **Barren** | 66 | 5 | 187 | 544 | 24 | 29 | 6 | 3882 | 3 | 13 | 4759 | 81.57 |
| **Water** | 2 | 6 | 9 | 4 | 0 | 239 | 1 | 25 | 2328 | 0 | 2614 | 89.06 |
| **Ice-Snow** | 0 | 1 | 1 | 33 | 0 | 0 | 0 | 5 | 1 | 1113 | 1154 | 96.45 |
| **Total** | 9564 | 19754 | 6567 | 8558 | 888 | 1762 | 962 | 4559 | 2380 | 1127 | 56121 | |
| **P.A.** | 88.27 | 92.85 | 62.30 | 67.97 | 90.65 | 53.69 | 93.76 | 85.15 | 97.82 | 98.76 | | |
| **O.A.** | | | | | | 83.16 | | | | | | |
| **Kappa** | | | | | | 0.789 | | | | | | |

439        To intuitively understand the spatial distribution of the GLC_FCS10 accuracy metrics, Fig. 5 presents the spatial
variations of O.A. and the kappa coefficient among 30 climate zones from Köppen climate zones. There is high
consistency of the O.A. and kappa coefficient within the spatial patterns, i.e., some climatic transition zones, land-
cover heterogeneity zones, cloud-contaminated tropical zones, and small subdivisions tend to have lower accuracy
(below 80%). Conversely, some homogeneous zones (such as Greenland and the Sahara Desert) and forest- or
cropland-rich zones (such as the east-central United States, central Eurasia, and East Asia) achieve the high O.A. and
kappa coefficient.

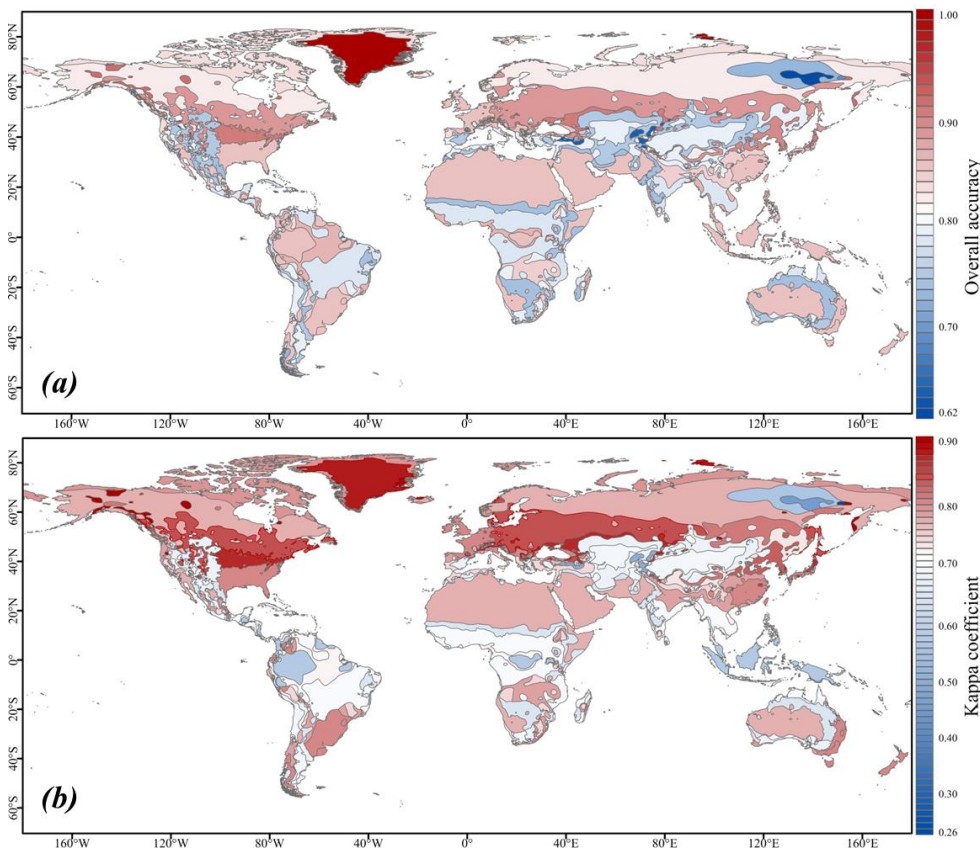


**Figure 5**. (a) Regional O.A. and (b) kappa coefficient of the GLC_FCS10 land-cover map among various Köppen
climate zones (http://koeppen-geiger.vu-wien.ac.at/) using the globally distributed validation points.

Table 4 further analyzes the confusion matrix between GLC_FCS10 and the global validation dataset with 16
land-cover types (refining the forest and cropland subcategories). Under this fine validation system, the GLC_FCS10
achieves an O.A. of 76.45% and a kappa coefficient of 0.736, which are reduced by 6.71% and 0.053, respectively,
from the metrics in Table 3. This reduction is due to confusion between the finer land-cover subcategories; e.g.,
forests have a U.A. value of 91.02% in Table 3, but when broken down into five forest subcategories the average
U.A. value drops to 68.52%. Taking mixed forest as an example, it has a low accuracy of only 44.17%, of which the
proportions misclassified as evergreen broadleaved forest (EBF), deciduous broadleaved forest (DBF), evergreen
needleleaved forest (ENF), and deciduous needleleaved forest (DNF) are 5.34%, 7.77%, 27.18%, and 5.83%,
respectively. Higher likelihoods of confusion exist for closely related land-cover subcategories; e.g., the highest
proportion of misclassification in EBF is in DBF, and rainfed cropland is easily misclassified as irrigated cropland,
shrubland, or grassland. Previous research also demonstrated that considerable misclassifications occur between
similar land-cover types (Homer et al., 2020; Wickham et al., 2021; Zhang et al., 2021; Zhang et al., 2024c).

**Table 4**. The confusion matrix between GLC_FCS10 and the globally distributed validation dataset for 16 land-cover
types.

| | RCP | ICP | EBF | DBF | ENF | DNF | MF | SHB | GRS | LMS | SPV | WET | IMP | BAL | WTR | SNW | Total | U.A. |
|---|---|---|---|---|---|---|---|---|---|---|---|---|---|---|---|---|---|---|
| **RCP** | 7575 | 302 | 134 | 137 | 21 | 13 | 6 | 574 | 746 | 0 | 10 | 24 | 31 | 35 | 6 | 0 | 9614 | 78.79 |
| **ICP** | 121 | 444 | 19 | 7 | 2 | 0 | 0 | 14 | 22 | 0 | 1 | 29 | 8 | 0 | 3 | 0 | 670 | 66.27 |
| **EBF** | 138 | 0 | 7263 | 842 | 289 | 59 | 48 | 429 | 70 | 0 | 0 | 140 | 1 | 1 | 2 | 0 | 9282 | 78.25 |
| **DBF** | 21 | 1 | 375 | 3955 | 235 | 149 | 61 | 482 | 52 | 0 | 0 | 40 | 0 | 1 | 0 | 0 | 5372 | 73.62 |
| **ENF** | 0 | 0 | 191 | 129 | 2535 | 533 | 11 | 138 | 28 | 0 | 0 | 51 | 1 | 3 | 1 | 0 | 3621 | 70.01 |
| **DNF** | 1 | 0 | 0 | 77 | 122 | 1278 | 4 | 127 | 37 | 0 | 1 | 16 | 0 | 6 | 1 | 0 | 1670 | 76.53 |
| **MF** | 0 | 0 | 11 | 16 | 56 | 12 | 91 | 15 | 2 | 0 | 0 | 3 | 0 | 0 | 0 | 0 | 206 | 44.17 |
| **SHB** | 184 | 6 | 60 | 440 | 52 | 140 | 9 | 4091 | 922 | 12 | 40 | 72 | 4 | 69 | 1 | 0 | 6102 | 67.04 |
| **GRS** | 654 | 19 | 24 | 109 | 21 | 98 | 3 | 275 | 5817 | 33 | 232 | 170 | 7 | 159 | 4 | 1 | 7626 | 76.28 |
| **LMS** | 0 | 0 | 0 | 6 | 1 | 18 | 0 | 78 | 153 | 805 | 18 | 2 | 0 | 43 | 0 | 0 | 1124 | 71.62 |
| **SPV** | 40 | 3 | 0 | 2 | 1 | 2 | 0 | 0 | 0 | 11 | 266 | 6 | 1 | 0 | 1 | 3 | 336 | 79.17 |
| **WET** | 8 | 2 | 29 | 9 | 9 | 27 | 0 | 136 | 100 | 14 | 12 | 946 | 1 | 11 | 30 | 0 | 1334 | 70.91 |
| **IMP** | 14 | 6 | 4 | 1 | 0 | 1 | 0 | 11 | 28 | 0 | 0 | 1 | 902 | 5 | 0 | 0 | 973 | 92.70 |
| **BAL** | 20 | 3 | 0 | 0 | 0 | 0 | 0 | 187 | 544 | 13 | 118 | 23 | 5 | 3498 | 2 | 10 | 4423 | 79.09 |
| **WTR** | 2 | 0 | 1 | 1 | 1 | 1 | 2 | 9 | 4 | 0 | 0 | 239 | 1 | 25 | 2328 | 0 | 2614 | 89.06 |
| **SNW** | 0 | 0 | 0 | 1 | 0 | 0 | 0 | 1 | 33 | 0 | 0 | 0 | 0 | 5 | 1 | 1113 | 1154 | 96.45 |
| **Total** | 8778 | 786 | 8111 | 5732 | 3345 | 2331 | 235 | 6567 | 8558 | 888 | 698 | 1762 | 962 | 3861 | 2380 | 1127 | 56121 | |
| **P.A.** | 86.30 | 56.49 | 89.55 | 69.00 | 75.78 | 54.83 | 38.72 | 62.30 | 67.97 | 90.65 | 38.11 | 53.69 | 93.76 | 90.60 | 97.82 | 98.76 | | |
| **O.A.** | | | | | | | | | 76.45 | | | | | | | | | |
| **Kappa** | | | | | | | | | 0.736 | | | | | | | | | |

**Note:** RCP: rainfed cropland, ICP: irrigated cropland, EBF: evergreen broadleaved forest, DBF: deciduous broadleaved forest, ENF: evergreen needleleaved forest, DNF: deciduous needleleaved forest, MF: mixed forest, SHB: shrubland, GRS: grassland, LMS: lichens and mosses, SPV: sparse vegetation, WET: wetland, IMP: impervious surface, BAL: bare areas, WTR: water body, SNW: permanent ice and snow

### 4.2.2 National-scale accuracy analysis using the LCMAP_Val datasets

Table 5 presents the confusion matrix for GLC_FCS10 based on the LCMAP_Val validation points over the
America. It should be noted that the LCMAP_Val dataset only contains eight land-cover types and merges shrubland
and grassland into one mosaiced land-cover type (grass/shrub). The GLC_FCS10 achieves O.A. of 85.09% and a
kappa coefficient of 0.804 using these 16082 national validation points. Regarding the U.A. and P.A., the cropland,

forest, water. and grass/shrub land-cover types achieve balanced U.A. and P.A. values approximating or exceeding
80%. In contrast, developed land has the lowest U.A. of 54.26% with high P.A. of 98.85%, mainly because of the
difference in definitions of developed land and impervious surfaces. The LCMAP_Val definition of developed land
is broad enough to classify inner-city greenery as developed land as well (Xian et al., 2022), which is considered a
vegetation land-cover type in the GLC_FCS10. Barren land has the lowest P.A. value of 31.93%, indicating a high
omission error of 68.07%. Most of these misclassifications came from the confusion between barren land and
grass/shrub land-cover types. It is noteworthy that the grass/shrub shares similar spectral characteristics with barren,
and both of them co-exist in arid regions of the western United States, thus, it is usually difficult to distinguish
between the two with high accuracy.

**Table 5.** The confusion matrix between GLC_FCS10 and the LCMAP_Val dataset.

| | Cropland | Forest | Grass/Shrub | Wetland | Impervious | Barren | Water | Ice & Snow | Total | U.A. |
|---|---|---|---|---|---|---|---|---|---|---|
| **Cropland** | 3445 | 28 | 393 | 6 | 0 | 9 | 2 | 0 | 3883 | 88.72 |
| **Forest** | 7 | 4621 | 133 | 92 | 0 | 0 | 2 | 0 | 4855 | 95.18 |
| **Grass/Shrub** | 368 | 358 | 3440 | 21 | 1 | 272 | 1 | 0 | 4461 | 77.11 |
| **Wetland** | 37 | 260 | 30 | 522 | 1 | 0 | 5 | 0 | 855 | 61.05 |
| **Developed** | 44 | 69 | 164 | 3 | 344 | 9 | 1 | 0 | 634 | 54.26 |
| **Barren** | 1 | 0 | 0 | 10 | 0 | 137 | 1 | 0 | 149 | 91.95 |
| **Water** | 0 | 2 | 1 | 63 | 2 | 1 | 1173 | 0 | 1242 | 94.44 |
| **Ice & Snow** | 0 | 0 | 0 | 0 | 0 | 1 | 0 | 2 | 3 | 66.67 |
| **Total** | 3902 | 5338 | 4161 | 717 | 348 | 429 | 1185 | 2 | 16082 | |
| **P.A.** | 88.29 | 86.57 | 82.67 | 72.80 | 98.85 | 31.93 | 98.99 | 100.00 | | |
| **O.A.** | | | | | 85.09 | | | | | |
| **Kappa** | | | | | 0.804 | | | | | |

Figure 6 illustrates the spatial distribution of O.A. and kappa coefficient values among different Köppen climate
zones using the LCMAP_Val validation points over the America. There is notable consistency between O.A. and
kappa coefficient in terms of the spatial patterns, i.e., the GLC_FCS10 considerably outperforms the Western U.S. in
the Eastern U.S. and has an optimal kappa coefficient of more than 0.8 in the Northeastern U.S. Combined with the
climatic distribution, it performs relatively poorly in the arid and semi-arid zones of the Midwestern U.S., mainly
attributed to the difficulty in distinguishing between shrubs, grasses, and bare land within the region.

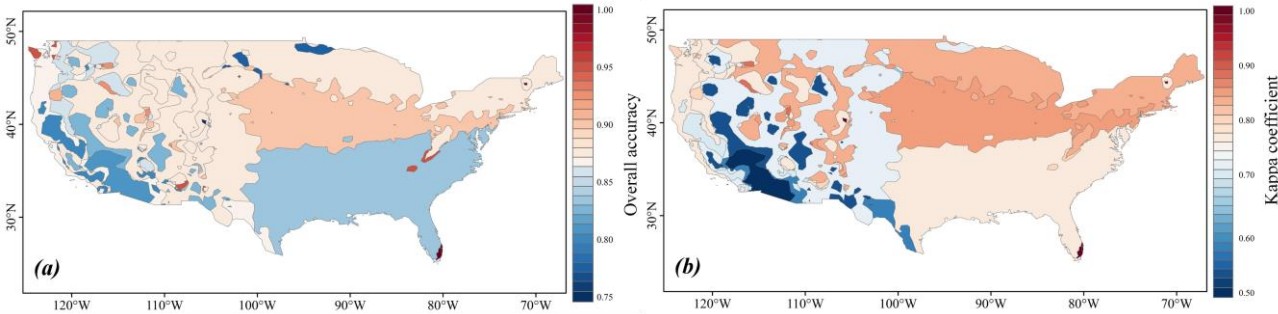

**Figure 6.** The spatial distributions of (a) O.A. and (b) kappa coefficient using the LCMAP_Val validation points over
the America among various Köppen climate zones.

### 4.3 Cross-comparisons with previous land-cover products

Table 6 gives quantitative comparisons among GLC_FCS10 and four public global 10- or 30-m land-cover
products using the LCMAP_Val dataset. The GLC_FCS10 achieves the highest O.A. of 85.09% and a kappa
coefficient of 0.804, followed by the ESA WorldCover (82.34% and 0.760) and ESRI_LC (82.10% and 0.754), while

the GLC_FCS30 and FROM_GLC10 have relatively inferior performance, below 80%. Specifically, in terms of the P.A., U.A. and F1-score, we can find that: 1) all five products achieve superior performance for water with corresponding P.A., U.A. and F1 score values of more than 90%. 2)The GLC_FCS10 and ESA WorldCover have advantages over other products for cropland and forest (with F1 scores exceeding 85%). The ESRI_LC F1 scores for cropland, forest, and grass/shrub exceed 80%. 3)All five products faced challenges for wetlands and barren land due to their complicated spectral characteristics. Taking wetlands as an example, the GLC_FCS10 achieves the highest F1 score of 66.63%, with most other products below 50%. The FROM_GLC10 has the lowest F1 score of 7.58%. 4) As stated in Section 4.2.2, the difference in definitions for developed land and impervious surfaces mean that all land-cover products have lower U.A. than P.A. values for developed land, i.e., they cannot identify inner-city greenery as impervious surfaces.

**Table 6.** Comparisons among GLC_FCS10 and four other comparative products using the LCMAP_Val dataset.

| | | Cropland | Forest | Grass/Shrub | Wetland | Developed | Barren | Water | Snow | O.A. | Kappa |
|---|---|---|---|---|---|---|---|---|---|---|---|
| **GLC_FCS10** | U.A. | 88.72 | 95.18 | 77.11 | 61.05 | 54.26 | 91.95 | 94.44 | 66.67 | | |
| | P.A. | 88.29 | 86.57 | 82.67 | 72.80 | 98.85 | 31.93 | 98.99 | 100.0 | 85.09 | 0.804 |
| | F1 | 88.50 | 90.67 | 79.79 | 66.41 | 70.06 | 47.40 | 96.66 | 80.00 | | |
| **FROM_GLC10** | U.A. | 71.14 | 87.13 | 73.70 | 4.02 | 41.64 | 90.50 | 96.03 | 100.0 | | |
| | P.A. | 89.17 | 82.42 | 73.47 | 66.00 | 85.31 | 9.30 | 98.34 | 50.00 | 74.31 | 0.653 |
| | F1 | 79.14 | 84.71 | 73.58 | 7.58 | 55.96 | 16.87 | 97.17 | 66.67 | | |
| **ESA WorldCover** | U.A. | 86.14 | 94.51 | 80.69 | 13.78 | 35.49 | 93.85 | 97.41 | 100.0 | | |
| | P.A. | 93.09 | 82.00 | 85.15 | 88.28 | 97.20 | 14.38 | 98.93 | 75.00 | 82.34 | 0.760 |
| | F1 | 89.48 | 87.81 | 82.86 | 23.84 | 52.00 | 24.94 | 98.16 | 85.71 | | |
| **ESRI_LC** | U.A. | 90.02 | 82.84 | 84.98 | 10.15 | 69.36 | 56.52 | 97.69 | 100.0 | | |
| | P.A. | 80.65 | 83.05 | 81.86 | 69.72 | 74.38 | 38.72 | 98.48 | 25.00 | 82.10 | 0.754 |
| | F1 | 85.08 | 82.94 | 83.39 | 17.72 | 71.78 | 45.96 | 98.08 | 40.00 | | |
| **GLC_FCS30** | U.A. | 85.78 | 88.85 | 75.29 | 37.68 | 38.10 | 73.49 | 90.79 | 100.0 | | |
| | P.A. | 77.39 | 74.69 | 84.47 | 56.09 | 91.70 | 20.08 | 98.14 | 75.00 | 77.76 | 0.699 |
| | F1 | 81.37 | 81.16 | 79.62 | 45.08 | 53.83 | 31.54 | 94.32 | 85.71 | | |

Figure 7 compares GLC_FCS10 with ESA WorldCover, ESRI_LC, GLC_FCS30, and FROM_GLC10 on the East Coast of the United States. Overall, there is the highest consistency between GLC_FCS10 and actual land-cover situations, i.e., wetlands are predominantly found in low-lying river valleys and along the coast, and with a cross-section of forests and cropland due to the undulating topography. Conversely, ESA WorldCover has the largest forest area because some swamps or woody wetlands are labeled as forests (Fig. 7R1 is an enlargement showing an example). ESA WorldCover also has the smallest impervious surface area because some is misclassified as forest (Fig. 7R2 is an enlargement showing an example). Thus, ESA WorldCover has low U.A. values of 13.78% and 35.49% for wetland and developed land, respectively (Table 6). ESRI_LC has the largest impervious surface area and also identifies some swamps as forest, so it has the lowest P.A. value of 74.38% for developed land. ESRI_LC overestimates impervious surfaces and has obvious omission errors for swamps. FROM_GLC10 has the lowest wetland area, i.e., some swamps are classified as forest and herbaceous wetlands are labeled as water, so it has the lowest U.A. value of 4.02% for wetlands in Table 6. Last, GLC_FCS30 also has omission errors for wetlands (the red rectangle on GLC_FCS30) and lacks spatial details for some small objects (such as small rivers in Fig. 7R1).

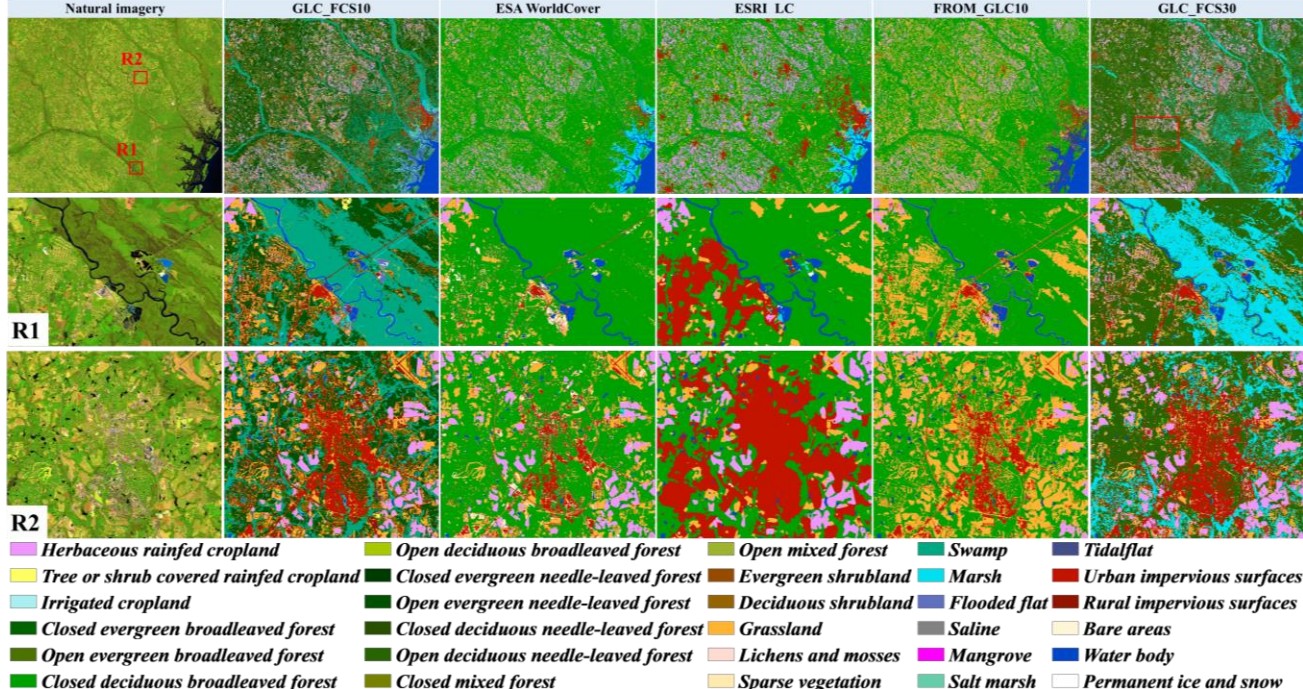

| Herbaceous rainfed cropland | Open deciduous broadleaved forest | Open mixed forest | Swamp | Tidalflat |
| Tree or shrub covered rainfed cropland | Closed evergreen needle-leaved forest | Evergreen shrubland | Marsh | Urban impervious surfaces |
| Irrigated cropland | Open evergreen needle-leaved forest | Deciduous shrubland | Flooded flat | Rural impervious surfaces |
| Closed evergreen broadleaved forest | Closed deciduous needle-leaved forest | Grassland | Saline | Bare areas |
| Open evergreen broadleaved forest | Open deciduous needle-leaved forest | Lichens and mosses | Mangrove | Water body |
| Closed deciduous broadleaved forest | Closed mixed forest | Sparse vegetation | Salt marsh | Permanent ice and snow |

**Figure 7**. Comparisons among GLC_FCS10 and ESA WorldCover, ESRI_LC, GLC_FCS30, and FROM_GLC10 on the East Coast of the United States. Images in the first column are false-color composited from time-series Sentinel-2 imagery.

Figure 8 presents comparisons for the moddle reaches of the Yangtze River, China. Overall, all land-cover products accurately capture the regional spatial patterns, and GLC_FCS10 and GLC_FCS30 have advantages with the diversity of land-cover types over the other three products. Specifically, Fig. 8R1 illustrates cross-comparisons for the megacity of Wuhan. ESA WorldCover underestimates and has the lowest impervious surface area, ESRI_LC overestimates and has the highest impervious surface area, and FROM_GLC10 misclassifies some impervious surfaces as grassland (Huang et al., 2022). Based on the former comparison and previous works (Huang et al., 2022), the ESA WorldCover underestimates these low-density impervious surfaces, the ESRI_LC suffers the overestimation problem on the impervious surfaces, and FROM_GLC10 suffers some misclassificaiton between impervious surfaces and grassland. Fig. 8R2 shows comparisons over the Payang Lake wetlands. ESA WorldCover captures most marsh wetlands but misses these lake/flooded flats, while ESRI_LC, FROM_GLC10, and GLC_FCS30 have serious omission errors for these wetlands. ESRI_LC misclassifies some marsh wetlands as grassland. Lastly, Figure 8R3 illustrates comparisons for mountainous areas, the ESRI_LC still overestimates impervious surfaces, and GLC_FCS30 misses some small impervious surface objects (roads) due to spatial resolution constraints.

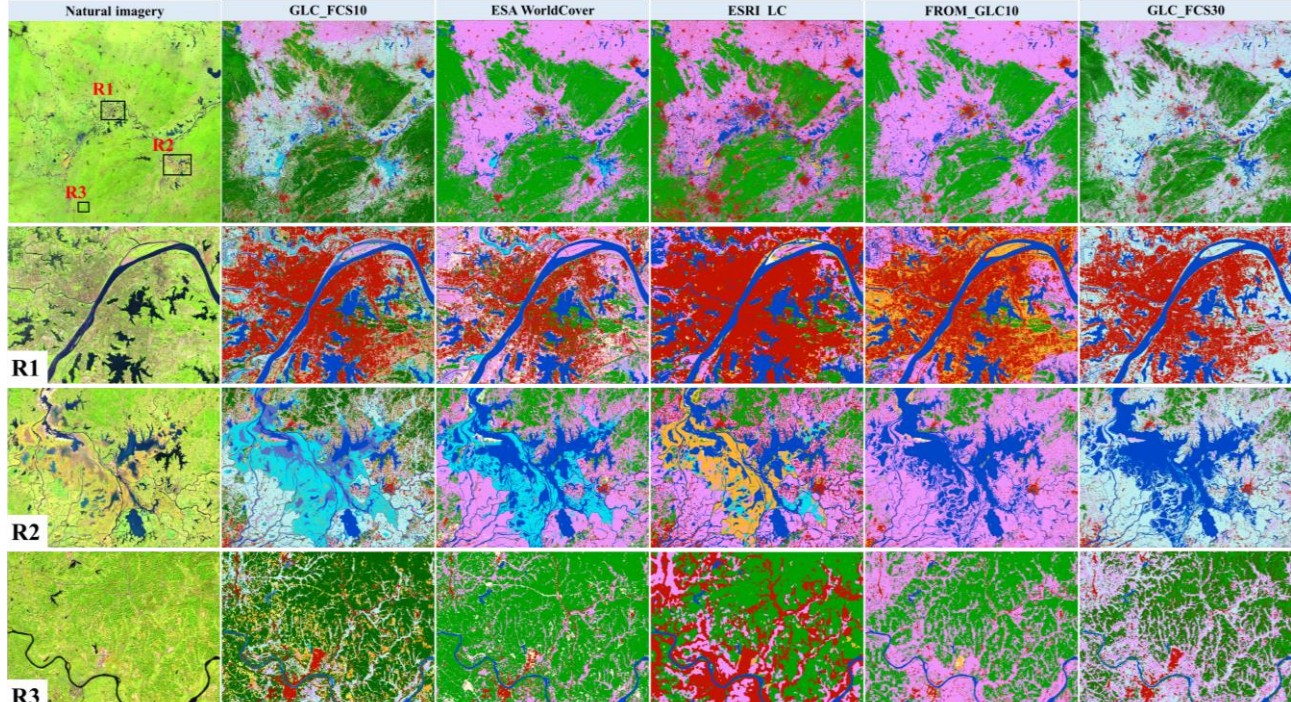

Figure 8. Comparisons among GLC_FCS10 and ESA WorldCover, ESRI_LC, GLC_FCS30, and FROM_GLC10 for the middle reaches of the Yangtze River, China. Images are derived from Sentinel-2 imagery using false-color compositing. The colormap of all land-cover products are same as in Fig. 6.

Since land-cover mapping usually meets great challenging on the tropic areas due to the frequent contaminations of cloud and shadow, Fig. 9 further shows comparisons for Kalimantan Island, Indonesia. It is noteworthy that the region has experienced extensive deforestation and oil palm cultivation over the past few decades (Descals et al., 2024). Overall, the GLC_FCS10, GLC_FCS30, and FROM_GLC10 can capture the spatial patterns of oil palms because they identify oil palms as cropland, while ESA WorldCover and ESRI_LC tend to treat oil palms as forest. Specifically, in the enlargement areas of Fig. 9R1, we can see more regular oil palm plantations due to human activities, while FROM_GLC10 and GLC_FCS30 might overestimate this oil palm croplands. Then, as for the local region R2 in which contains swamp, mangrove and oil palms, the ESRI_LC and FROM_GLC have serious omission errors on mangroves and swamps while ESA WorldCover still cannot identify these swamp wetlands and oil palms, and GLC_FCS30 is consistent with GLC_FCS10 in capturing the wetlands and oil palms.

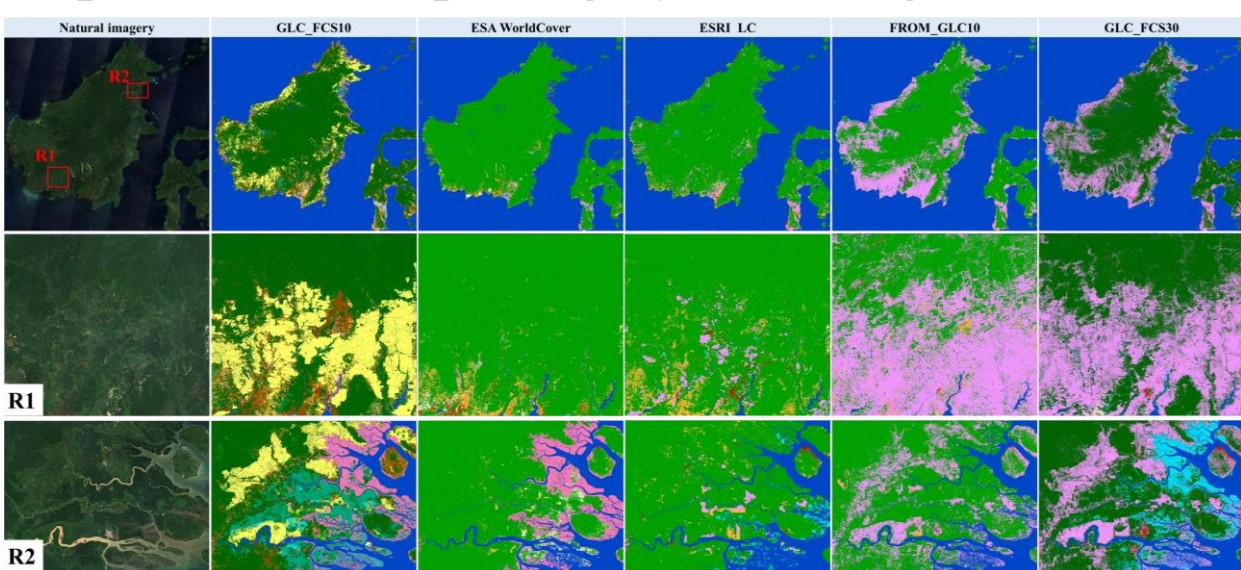

**Figure 9**. Comparisons between GLC_FCS10 and ESA WorldCover, ESRI_LC, GLC_FCS30, and FROM_GLC10 for Kalimantan Island, Indonesia. Images are derived from Sentinel-2 imagery in 2023, and the colormap is the same as in Fig. 7.

## 4.4 The feasibility and benefits of the proposed method for large-area land-cover mapping

### 4.4.1 The feasibility and advantages of globally derived training samples

A principal difficulty of land-cover mapping is obtaining high-quality training samples (Li et al., 2023; Zhang et al., 2021), in this work, we integrate prior multisource global land-cover products to generate globally distributed training samples. To ensure the confidence of these derived training samples and minimize the classification errors of each prior product, we took the following actions: 1) spatiotemporal consistency checking was used to find homogeneous and stable areas. 2) The intersection of multiple land-cover products minimized the influence of classification errors in each product. 3) A morphological erosion filter was applied to reduce the impact of edge-mixing effects. The accuracy assessment partly demonstrates the reliability of these derived training samples, i.e., GLC_FCS10 achieves satisfactory accuracy metrics and outperforms several other land-cover products. Due to the large volume of these globally distributed training samples, we selected approximately 10,000 derived samples from the training sample pool in Section 3.2.4. Upon meticulous inspection, we determined that these chosen samples attained an overall accuracy (O.A.) of 92.18%, with certain uncertainties existing for shrubland and grassland. This result was in accordance with the earlier analysis presented in Table 3.

Moreover, it is still uncertain whether this small amount of erroneous training samples could impact the performance of land-cover mapping, Fig. 10 illustrates the quantitative relationship between the erroneous training samples and the O.A. and kappa coefficients for the basic classification system. Initially, O.A. and the kappa coefficient remain stable as the number of erroneous training samples increases. However, a significant decline occurs when the proportion of erroneous samples exceeds 30%. This indicates that the trained random forest model is robust to the erroneous training samples as long as their proportion remains below 30%. In this work, if the fraction of erroneous samples was kept below 30%, the difference in O.A. is approximately 2% and the decrease in the kappa coefficient is approximately 3%. Gong et al. (2024) also demonstrated that a small number of incorrect samples (approximately 20%) didn't affect the land-cover classification accuracy.

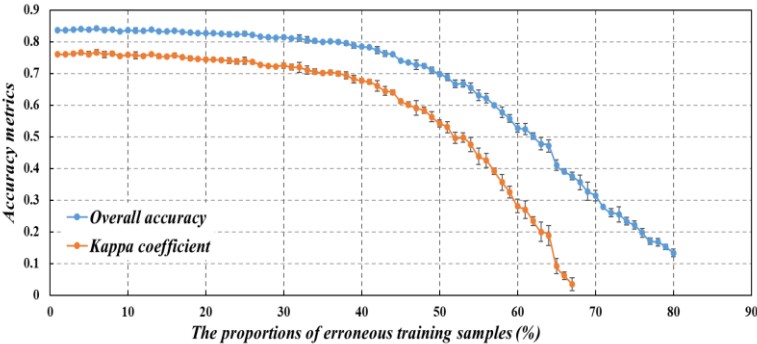

**Figure 10**. A sensitive analysis of kappa coefficient and O.A. with respect to the proportion of erroneous training samples.

### 4.4.2 The advantages of hierarchical land-cover mapping strategy

One of the novelties of this study is the adoption of the hierarchical land-cover mapping strategy. The accuracy assessment in Table 3 indicates that impervious surfaces have a high U.A. value of 92.70% and P.A. value of 93.76%. The wetlands U.A. and P.A. values are 70.91% and 53.69%, which were superior to those of the other land-cover products in Table 6 and the cross-comparisons in Figs. 7–9. To intuitively understand the advantage of the hierarchical land-cover mapping strategy, a comparative experiment (***ComExp***) has been designed using the training samples

from area-fraction allocation (explained in the Section 3.2.4)in the Fig. 11, i.e., the impervious surfaces and wetlands
are not classified separately in the middle reaches of the Yangtze River (the comparative site in Fig. 7). Overall, the
***ComExp*** is consistent with the GLC_FCS10 spatial patterns and shows some variations in the details. Specifically,
in the Fig. 10R1, the ***ComExp*** misclassifies some marsh wetlands as herbaceous rainfed cropland, i.e., some lake
flats (red rectangles in Fig. 11R1) cannot be comprehensively captured when compared with the GLC_FCS10. Figure
10R2 gives the comparisons on the impervious surface areas, we can find that the ***ComExp*** has lower impervious
surface areas because it misclassifies some bright impervious surfaces as bare areas and some residential areas as
vegetated land. In summary, the hierarchical land-cover mapping strategy can increase the ability to characterize
specific land-cover types. Similarly, the work of Sulla-Menashe et al. (2019) also used hierarchical mapping to
generate annual global land-cover types for MCD12Q1 and demonstrated its better performance.

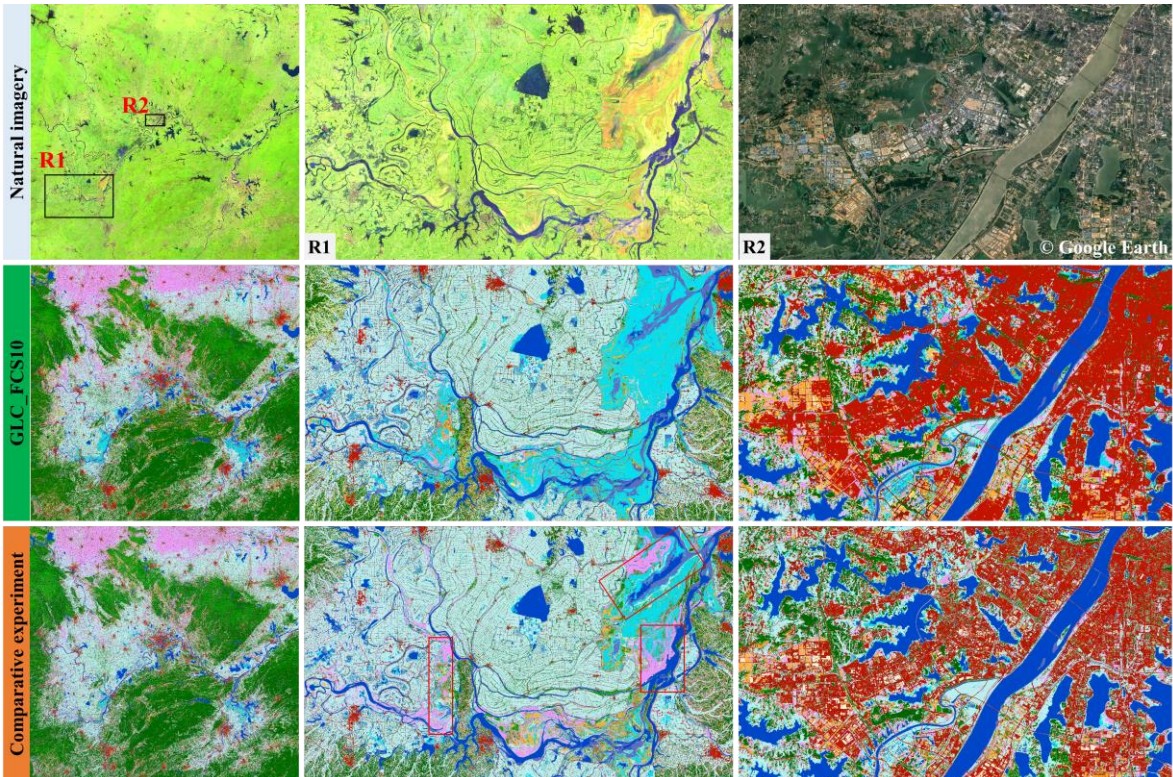


**Figure 11**. A comparative experiment on whether to adopt the hierarchical land-cover mapping strategy over the
middle reaches of the Yangtze River. Images are composited from Sentinel-2 imagery, and the enlargement came
from © Google Earth. The colormap is the same as in Fig. 7.

## 4.5 The limitations and prospects

By combining a globally distributed training sample pool and the hierarchical land-cover mapping strategy, a
novel GLC_FCS10 product containing 30 category-detailed land-cover types have been produced. GLC_FCS10
achieves more accurate performance than several previous products in quantitative and qualitative comparisons
(Sections 4.2 and 4.3). However, there are still some limitations or uncertainties regarding the proposed method and
the developed products, which will be addressed in our ongoing works. First, although we combined hierarchical
land-cover mapping and multisource satellite observations to improve the recognition of complicated land-cover
types (impervious surfaces and wetlands), however, it can be found that the accuracy metrics of shrubland, grassland
and wetland still have substantial room for improvement. Recently, some efforts have shown that incorporating both
climatic and landform factors can enhance the discrimination of grassland areas (Parente et al., 2024), and the
combination of Lidar (Light Detection And Ranging) and optical information increases the separability of shrubland
and forest (Prošek and Šímová, 2019). Thus, one of the further works will combine multisource information to

enhance the ability to recognize more complex land-cover types. Meanwhile, although the combination of time-series Sentinel-1 and Sentinel-2 can minimize the effect of clouds and shadows, some high cloud-contaminated areas might be still affected, i.e., mosaic seams may be visible in these special areas. Many previous studies have demonstrated that the harmonization of Landsat and Sentinel-2 can increase the likelihood of clear observations (Claverie et al., 2018), and the advances of deep learning models also improve the land-cover mapping performance on these cloudy areas (Xu et al., 2024a). Thus, how to make full use of the Landsat imagery and deep learning techniques to further improve the quality of GLC_FCS10 in the persistent cloudy areas will be one of the future works.

We collected a globally distributed validation dataset and one third-party validation dataset (LCMAP_Val) for the purpose of quantifying the performance of the GLC_FCS10. However, the accuracy metrics of GLC_FCS10 for the fine classification system (containing 30 land-cover types) is still unknown. Actually, some previous studies have emphasized that collecting a large-area validation dataset is quite challenging (Morales et al., 2023; Tsendbazar et al., 2021; Xu et al., 2020), especially as this study also needed to focus on 30 fine land-cover types. Fortunately, over the past decades, many previous works have collected high-quality validation points at global or regional scales (d'Andrimont et al., 2020; Li et al., 2017; Stanimirova et al., 2023; Stehman et al., 2012; Zhao et al., 2023). Making full use of these prior knowledge bases to refine the globally distributed validation points into 30 fine land-cover types will be another focus for ongoing work. In addition, to objectively understand the accuracy performance of GLC_FCS10, we introduced the LCMAP_Val third-party validation dataset, but the differences in the definition of the classification system still affect the accuracy metrics, such as the higher P.A. and the lower U.A. for the impervious surfaces in Table 5. Therefore, one of the ongoing works would take some measures (such as: semantic similarity (Gao et al., 2020)) for more comprehensively and objectively assessing the third-party accuracy metrics of GLC_FCS10.

Lastly, in order to maximize the utilization of training samples distributed worldwide and strengthen the classification modeling capacity for capturing regional characteristics, the local adaptive modeling strategy (Section 3.4) was applied in each 5 × 5 geographical tile, i.e., the global land-cover maps were produced as 984 independent local adaptive models. There may be a slight spatial discontinuity in some local land-cover maps between neighboring areas even though we introduced spatial neighborhood information into the regional modeling. Thus, further work will take some measures to join global and regional sample modeling to enhance the spatial continuity of global land-cover maps.

## 5. Data availability

In this study, the new GLC_FCS10 land-cover dataset with the fine classification system in 2023 has been uploaded to the Zenodo platform and can be visually visited at https://zhangxiao-glcproj.users.earthengine.app/view/glcfcs102023maps and freely access at https://doi.org/10.5281/zenodo.14729665 (Liu and Zhang, 2025). To facilitate the use of this dataset, the global GLC_FCS10 dataset has been stored by a total of 984 independent 5 × 5 geographical tiles, and the tile names as "GLC_FCS10_2023_E/W***N/S##," in which the "***" and "##" illustrate coordinates of longitude and latitude at the upper - left corner of the tile data.

As the collection of global validation dataset is labor intensive and time-consuming, our globally distributed validation dataset in 2023 will be available upon reasonable request.

## 6. Conclusion

The continuous improvement of satellite techniques and computational capability provide ample opportunity for high-resolution global land-cover mapping. In this work, we proposed a framework that leverages prior multisource

land-cover products, hierarchical land-cover mapping, and local adaptive classification to generate a novel GLC_FCS10 global land-cover product containing 30 fine land-cover types in 2023 from time-series Sentinel-1 and Sentinel-2 imagery on the GEE platform. The GLC_FCS10 was validated to achieve an O.A. of 83.16% and a kappa coefficient of 0.789 using 56121 globally distributed validation points and achieved an O.A. of 85.09% in the United States using a third-party validation dataset. Furthermore, cross-comparisons with several public global high-resolution land-cover products also demonstrated that GLC_FCS10 had advantages on the diversity of land-cover types and capturing spatial details. Therefore, the GLC_FCS10 is a novel global 10-m land-cover product with high accuracy and a fine classification system. It can provide vital support for high-resolution land-cover applications.

## Acknowledgment

We extend our sincere gratitude to all the data providers and organizations whose datasets have been utilized in our research paper, the cloud computing power provided by Google Earth Engine platform, and the free storage services offered by the Zenodo platform.

## Author contributions

The concept and investigation of the project were carried out by LL and XZ. XZ and XC devised the research methodology. TZ, WZ, and MB were responsible for conducting the validation. XZ took the lead in drafting the initial version of the paper. LL and LG engaged in the review and editing process of the paper.

## Financial support

This research was supported by the National Key Research and Development Program of China (2024YFF0808301) and the National Natural Science Foundation of China (42201499).

## Competing interests

The corresponding author has declared that none of the authors have any competing interests.

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
