# Peer review of "GLC FCS10: a global 10-m land-cover dataset with a fine classification system from Sentinel-1"

_Earth System Science Data, 2025_

## Author Response (AR1)

Dear Topical Editor and Reviewers:

On behalf of my co-authors, we thank you very much for reviewing our manuscript and giving us a lot of useful comments and suggestions. We appreciate the comments on our manuscript entitled "*GLC_FCS10: a global 10-m land-cover dataset with a fine classification system from Sentinel-1 and Sentinel-2 time-series data in Google Earth Engine*" (essd-2025-73).

We have revised the manuscript carefully according to the comments. All the changes were high-lighted (red color) in the manuscript. And the point-by-point response to the comments of the reviewers is also listed below.

Looking forward to hearing from you soon.

Best regards,

Prof. Liangyun Liu liuly@radi.ac.cn

Institute of Remote Sensing and Digital Earth, Chinese Academy of Sciences

No.9 Dengzhuang South Road, Haidian District, Beijing 100094, China

**Response to comments**
**Paper #:** essd-2025-73
**Title:** GLC_FCS10: a global 10-m land-cover dataset with a fine classification system from Sentinel-1 and Sentinel-2 time-series data in Google Earth Engine
**Journal**: Earth System Science Data

**Reviewer #1**

This work is of great importance in multiple fields, such as earth science, geography, terrestrial ecosystem. These dataset could be very useful in many places. But some shortcomings in methods, figures, and tables need careful clarification and revisement, here are comments:

Great thanks for the positive comments. The manuscript has been greatly improved based on your and another reviewers' comments and suggestions.

1. Fig 1 is too colorful to demonstrate the core idea. Also, there is no legend to show what are different colors indicating. The relationships between different boxes are confusing too. Suggesting simplifying the figure by summarizing the main and key steps using icon and/or key words, do not put every step and all the datasets in this single figure; be consistent in color, size, and font. More detailed techniques can be in new figure.

Great thanks for the comment. Based on the suggestion, the flowchart of the proposed method has been simplified as:

[Figure]

**Figure 1**. The flowchart of the proposed method for hierarchical land-cover mapping.

2. Table 2. how was the newly added forest- and wetland-related subcategories defined in the new system? What are the quantitative standards for closed vs open forest? Need more justification. These subclasses will of great importance in understanding the diversity of forest and wetland ecosystem, but only if the definition and classification of these subclasses are reasonable and practical.

Great thanks for the comment. Based on your constructive suggestion, the descriptions about the fine classification system have been added in the Table S1 as:

**Table S1**. The detailed definitions of 30 land-cover types in the fine classification system.

| Fine classification system | Definition |
| --- | --- |
| Herbaceous rainfed cropland | Herbaceous cropland with no irrigation facilities and crops grown by natural precipitation |
| Tree or shrub covered rainfed cropland (orchard, oil palm…) | Tree or shrub covered rainfed cropland, mainly including orchard, oil palm, etc. |

| Irrigated cropland | Cropland with guaranteed water sources and irrigation facilities that can be irrigated normally in a typical year |
|---|---|
| Closed evergreen broadleaved forest | Evergreen broadleaved tree cover, tree height > 3 m, tree-cover percentage > 40% |
| Open evergreen broadleaved forest | Evergreen broadleaved tree cover, tree height > 3 m, 15%< tree-cover percentage < 40% |
| Closed deciduous broadleaved forest | Deciduous broadleaved tree cover, tree height > 3 m, tree-cover percentage > 40% |
| Open deciduous broadleaved forest | Deciduous broadleaved tree cover, tree height > 3 m, 15%< tree-cover percentage < 40% |
| Closed evergreen needleleaved forest | Evergreen needleleaved tree cover, tree height > 3 m, tree-cover percentage > 40% |
| Open evergreen needleleaved forest | Evergreen needleleaved tree cover, tree height > 3 m, 15%< tree-cover percentage < 40% |
| Closed deciduous needleleaved forest | Deciduous needleleaved tree cover, tree height > 3 m, tree-cover percentage > 40% |
| Open deciduous needleleaved forest | Deciduous needleleaved tree cover, tree height > 3 m, 15%< tree-cover percentage < 40% |
| Closed mixed-leaf forest | Mixed broadleaved and needleleaved forests, tree height > 3 m, tree-cover percentage > 40% |
| Open mixed-leaf forest | Mixed broadleaved and needleleaved forests, tree height > 3 m, 15%< tree-cover percentage < 40% |
| Evergreen shrubland | Vegetation communities dominated by low cover and evergreen dwarf and scrub woodlands |
| Deciduous shrubland | Vegetation communities dominated by woody shrubs that lose their leaves in winter or the dry season |
| Grassland | Refers to land where herbaceous plants predominate |
| Lichens and mosses | Lichens and moss-covered areas |
| Swamp | The forest or shrubs which grow in the inland freshwater |
| Marsh | Herbaceous vegetation (grasses, herbs and low shrubs) grows in the freshwater |
| Lake/river flat | The non-vegetated flooded areas along the rivers and lakes |
| Saline | Characterized by saline soils and halophytic (salt tolerant) plant species along saline lakes |
| Mangrove forest | The forest or shrubs which grow in the coastal brackish or saline water |
| Salt marsh | Herbaceous vegetation (grasses, herbs and low shrubs) in the upper coastal intertidal zone |
| Tidal flat | The tidal flooded zones between the coastal high and low tide levels including mudflats and sandflats |
| Urban impervious surfaces | Land covered with buildings and other man-made structures within the urban boundary |
| Rural impervious surfaces | Land covered with man-made structures outside the urban boundary, mainly including rural residential land, transportation land, etc. |
| Sparse vegetation | Areas covered by woodland, shrubs and grasses, vegetation-cover percentage < 15% |
| Bare areas | Refers to land that is largely devoid of vegetation cover |
| Water | Lakes, rivers and streams that are always flooded |
| Permanent ice and snow | Areas covered by snow and ice all year round |

3.    How was the GLC_FCS30D dataset used as the training dataset for the 10m global cover mapping in this study? They are in different spatial resolutions, also, there are uncertainties in the GLC_FCS30D, not to mention the GLC_FCS30D does not cover 2023. How were all the uncertainties in the training dataset evaluated? Without solid evaluation, these training dataset cannot be high-confidence.

Great thanks for the comment. Yes, we completely agree that the high-confidence of training dataset is the key of subsequent land-cover mapping. In this study, three measures were used to identify these spatiotemporal homogeneity and high-quality training areas from GLC_FCS30D: 1) A time-series consistency analysis is applied to the GLC_FCS30D, and only stable areas during 1985–2022 will be retained as $TrainCanArea\_NLCs$. 2) The $MaxBound_{imp}$ and $MaxBound_{wet}$ are imported to mask the $TrainCanArea\_NLCs$, i.e., the training areas for non-wetland natural land-cover types should be located outside of $MaxBound_{imp}$ and $MaxBound_{wet}$. The aim of this step is to minimize confusion between non-wetland natural land-cover types and these two land-cover types. 3) A morphological erosion filter with a local window of 3 pixels × 3 pixels is used to find the spatially homogeneous areas for non-wetland natural land-cover types.

In terms of the different spatial resolutions between GLC_FCS30D with the need of 10 m training samples, the **"metric centroid"** method is adopted, which had been used to downscale 500-m training samples from MCD12Q1 to 30-m in the work of Zhang and Roy (2017). The detailed descriptions about the **"metric centroid"** method have been explained as:

Second, most high-quality training samples (except for those for impervious surfaces) are derived from the 30-m training areas, so there is also a need to reduce the 30-m training samples to 10-m samples to achieve a global 10-m land-cover map. In this work, the "metric centroid" method is adopted, which had been used to downscale 500-m training samples from MCD12Q1 to 30-m in the work of Zhang and Roy (2017). Specifically, as each 30-m pixel corresponds to 3 × 3 10-m pixels, we first find the centroid from these nine pixels as $P_{centroid}$ through spectral averaging, and then the point with the smallest absolute distance with $P_{centroid}$ was chosen as the optimal downscaled 10-m sample point [**Eq. (4)**].

$$P_i = \underset{i}{\arg\min}\left(\left|\boldsymbol{\rho}_{P_i} - \boldsymbol{\rho}_{P_{centroid}}\right|\right), \boldsymbol{\rho}_{P_{centroid}} = \frac{1}{9}\sum_{j=1}^{9}\frac{\boldsymbol{\rho}_{P_j}}{9} \tag{4}$$

Where $\boldsymbol{\rho}_{P_i}$ is the spectra value of composited Sentinel-2 training features (See Section 3.3) at pixel $P_i$. If more than one point in the nine pixels has the same minimum absolute distance, then we pick randomly from among them.

Then, as for the uncertainties in these derived training samples, due to the large volume of these globally distributed training samples, we selected approximately 10,000 derived samples from the training sample pool. Upon meticulous inspection, we determined that these chosen samples attained an overall accuracy (O.A.) of 92.18%, with certain uncertainties existing for shrubland and grassland. It has been explained in the Section 4.4.1 as:

A principal difficulty of land-cover mapping is obtaining high-quality training samples (Li et al., 2023; Zhang et al., 2021), in this work, we integrate prior multisource global land-cover products to generate globally distributed training samples. To ensure the confidence of these derived training samples and minimize the classification errors of each prior product, we took the following actions: spatiotemporal consistency checking was used to find homogeneous and stable areas. The intersection of multiple land-cover products minimized the influence of classification errors in each product. A morphological erosion filter was applied to reduce the impact of edge-mixing effects. The accuracy assessment partly demonstrates the reliability of these derived training samples, i.e., GLC_FCS10 achieves satisfactory accuracy metrics and outperforms several other land-cover products. Similarly, Zhang et al. (2021) also used the prior global land-cover products to generate the GLC_FCS30 product with satisfactory performance. Due to the large volume of these globally distributed training samples, we selected approximately 10,000 derived samples from the training sample pool in Section 3.2.4. Upon meticulous inspection, we determined that these chosen samples attained an overall accuracy (O.A.) of 92.18%, with certain uncertainties existing for shrubland and grassland. This result was in accordance with the earlier analysis presented in Table 3.

In addition, as we all know that automatically derived training samples cannot be guaranteed to be completely accurate, so whether these erroneous training samples affect the subsequent mapping accuracy. Therefore, this manuscript further discusses the quantitative relationship between the proportion of erroneous training samples and overall classification accuracy in Section 4.4.1 as:

Moreover, it is still uncertain whether this small amount of erroneous training samples could impact the performance of land-cover mapping, Fig. 9 illustrates the quantitative relationship between the erroneous training samples and the O.A. and kappa coefficients for the basic classification system. Initially, O.A. and the kappa coefficient remain stable as the number of erroneous training samples increases. However, a significant decline occurs when the proportion of erroneous samples exceeds 30%. This indicates that the trained random forest model is robust to the erroneous training samples as long as their proportion remains below 30%. In this work, if the fraction of erroneous samples was kept below 30%, the difference in O.A. is approximately 2% and the decrease in the kappa coefficient is approximately 3%. Gong et al. (2024) also demonstrated that a small number of incorrect samples (approximately 20%) didn't affect the land-cover classification accuracy.

[Figure]

**Figure 9**. A sensitive analysis of kappa coefficient and O.A. with respect to the proportion of erroneous training samples.

Lastly, as for the GLC_FCS30D does not cover 2023, yes, the released GLC_FCS30D only covered the period of 1985-2022, actually, the land cover change mask between 2022-2023 has also been produced using the continuous change detection method, but not yet shared with the public, and that change mask dataset was also used to generate the training samples in this manuscript to ensure the temporal consistency between GLC_FCS30D and training samples. It has been added in the Section 3.2.3 as:

**In addition, it should be noted that the TrainCanArea_NLCs represents the stable areas during 1985-2022, and there is still one-year interval with the land-cover mapping year in 2023. Fortunately, the ongoing updating of GLC_FCS30D is still in progress, the land-cover change masks during 2022-2023 have been finished, and which are also used to remove these changed and low-confidence areas.**

4.    Line 277, how as the percentiles be quantified, by date? by quality or what? So was the VV and VH percentiles in Line 280.

Great thanks for the comment. The quantile-based compositing method rearranges intra-annual time-series reflectance according to **mathematical magnitude and take the corresponding quartiles to reflect the phenological variation of the time-series.** Figure S1 illustrates the schematic of how to extract quantile features from time-series satellite observations, and we can find that these percentiles can efficiently capture the variations of phenology.

[Figure]

Figure S1. The schematic of how to extract quantile features from time-series satellite observations.

The basic principle of the quantile-based compositing method has been added as:

**The basic principle of this method is to rearrange intra-annual time-series reflectance according to mathematical magnitude and take the corresponding quartiles to reflect the phenological variation of the time-**

**series and suppress the noise interference such as clouds and shadows (Hansen et al., 2014). Some previous studies have also demonstrated its ability to flexibly balance noise removal and signal retention, reflecting the surface normality and capturing peak features (e.g., high percentile), adapting to different monitoring needs (Hansen et al., 2014; Zhang and Roy, 2017).** Thus, in this study, time-series Sentinel-2 images are composited into the 10th, 30th, 50th, 70th, and 90th percentiles for their 10 optical bands from visible to shortwave infrared and three typical indexes [NDVI, NDWI, and LSWI in **Eq. (5)**] using the percentile-based compositing method.

5.    Line 297-300 add a figure to show how was the hierarchical land cover constructed.

Great thanks for the comment. Based on your suggestion, the flowchart of how to build the hierarchical land-cover mapping models has been added as:

[Figure]

Figure 2. The detailed flowchart of hierarchical land-cover mapping algorithm by integrating globally distributed training samples and multisourced composited features.

6.    Line 307 what are the 5 × 5 geographical tiles indicating? how large is the tile, and why choosing 5×5?

Great thanks for pointing out the issue. The $5° × 5°$ geographical tile is a regional unit for local adaptive modeling, and the size of each $5° × 5°$ geographical tile is equivalent to **556 km × 556 km on the equator**, and the Figure S2 gives the overview of these 983 $5° × 5°$ geographical tiles used for local adaptive modeling.

In terms of why we choose these $5° × 5°$ geographical tiles, the reasons are concluded as: 1) the local adaptive modeling strategy has been demonstrated to achieve the superior performance than the single land-cover global modeling; 2) the previous works of Zhang et al., (2017) and Zhang et al., (2019) have explained that the training samples of sparse land-cover types in a small geographical grid were usually missed or greatly sparse, and the training samples from neighboring 3-by-3 tiles were also imported; 3) the GEE platform also had some limitations for computation capacity and memory; 4) the previous works in global 30 m land-cover mapping and change monitoring also demonstrated the efficiency and accuracy of these $5° × 5°$ geographical tiles. In summary, after balancing the accuracy performance, computation efficiency, and training sample volume, the $5° × 5°$ geographical tile is used to train the local adaptive classification models.

[Figure]

Figure S2. Overview of the 5° × 5° geographical tiles used for local adaptive modeling; the globe land area was split into 983 5° ×5° geographical tiles.

In the revised manuscript, the explanation of why we choose these 5° ×5° geographical tiles have been added as:

Then, **we split the globe into 984 5° × 5° geographical tiles (approximately 556 km × 556 km on the equator, illustrating on the Figure S1), because some studies emphasized that the local adaptive modeling usually achieves better mapping accuracy than single land-cover global modeling (Zhang et al., 2021), and previous works of Zhang and Roy (2017) and Zhang et al. (2019) have explained that the training samples of sparse land-cover types in a small geographical grid were usually missed or greatly sparse. Thus, after balancing the training sample volume, mapping accuracy, and the limitation of GEE platform, the local modeling tile size of 5° ×5° , similar to the works of (Zhang et al., 2021; Zhang et al., 2024), were used.** Then, when building the training model for each 5° × 5° geographical tile, we also import training samples within their spatial neighborhood of 3 × 3 tiles to ensure spatial consistency over the adjacent tiles.

7. Fig 3 is a map not maps.

Great thanks for the comment. It has been revised over the manuscript.

8. Table 3. why only 10 types evaluated, what about the other 20 types?

Great thanks for the comment. To comprehensively evaluate the accuracy metrics of GLC_FCS10, we designed two level of validation system (containing various land-cover details).

First, the **10 major land-cover types** correspond to the basic classification system, we can find a mapping relationship between the 30 fine types and the 10 base types in the Table 2, i.e., **the 30 fine land-cover types are merged into 10 major land-cover types in the Table 3**. The aim of Table 3 is to give the overall accuracy metrics under basic classification system.

At present, based only on current visual interpretation means and information, we can't decipher the global validation sample points for all 30 fine land-cover categories, and therefore, the confusion matrix under the 30 types cannot be given at this time. Thus, this limitation has been discussed on our Section 4.5 as:

We collected a globally distributed validation dataset and one third-party validation dataset (LCMAP_Val) for the purpose of quantifying the performance of the GLC_FCS10. However, the accuracy metrics of GLC_FCS10 for the fine classification system (containing 30 land-cover types) is still unknown. Actually, some previous studies have emphasized that collecting a large-area validation dataset is quite challenging (Tsendbazar et al., 2021; Xu et al., 2020), especially as this study also needed to focus on 30 fine land-cover types. Fortunately, over the past decades, many previous works have collected high-quality validation points at global or regional scales (d'Andrimont et al., 2020; Li et al., 2017; Stanimirova et al., 2023; Stehman et al., 2012; Zhao et al., 2023). Making full use of these prior knowledge bases to refine the globally distributed validation points into 30 fine land-cover types will be another focus for ongoing work.

what are the bottom line of OA. Kappa mean, why they are different from others?

To the best of our ability, we comprehensively evaluated the performance of GLC_FCS10 at different levels (i.e., merged into different class systems), utilizing both global and third-party regional validation sample data. The O.A. and Kappa in Table 3 give the overall accuracy metrics under basic classification system, i.e., the GLC_FCS10 has been transformed into basic classification system (10 basic land-cover types).

The reasons why they are different from the metrics in the Table 4~6 are: 1) the Table 3 and Table 4 correspond to different validation systems, i.e., Table 4 further refines forest and cropland; 2) Table 3 applies a different validation data source than the subsequent Tables 5&6, with the latter applying the third-party LCMAP_Val dataset.

**Response to comments**
**Paper #:** essd-2025-73
**Title:** GLC_FCS10: a global 10-m land-cover dataset with a fine classification system from Sentinel-1 and Sentinel-2 time-series data in Google Earth Engine
**Journal**: Earth System Science Data

**Reviewer #2**

This study developed a novel global 10-m land-cover dataset with a fine classification system. The data performs well and is of great value to high-resolution land-cover applications. Here are some of my concerns:

Great thanks for your positive comment, the manuscript has been further improved based on your and another reviewer's constructive and useful comments.

1. As highlighted in previous studies (Wang et al, 2023; Xu et al, 2024), Imp-ESRI_LC exhibits extensive patches of the impervious surface and lacks spatial details, which can also be found in Figure 6. This raises concerns about the MaxBound_imp (the union of several impervious surface products), which appears to include substantial areas of inner-city vegetation. If all training samples for natural land cover types are collected outside the MaxBound_imp, could this lead to the omission of inner-city vegetation types like grass and avenue trees?
See: Wang, Y., Xu, Y., Xu, X., Jiang, X., Mo, Y., Cui, H., Zhu, S., and Wu, H.: Evaluation of six global high- resolution global land cover products over China, International Journal of Digital Earth.
Xu, P., Tsendbazar, N.-E., Herold, M., de Bruin, S., Koopmans, M., Birch, T., Carter, S., Fritz, S., Lesiv, M., Mazur, E., Pickens, A., Potapov, P., Stolle, F., Tyukavina, A., Van De Kerchove, R., and Zanaga, D.: Comparative validation of recent 10 m-resolution global land cover maps, Remote Sensing of Environment.

Great thanks for the comment. Yes, we agree that the Imp-ESRI_LC exhibits extensive patches of the impervious surface and lacks spatial details. In terms of the concerns about the MaxBound_imp affects the omission error of inner-city vegetation types like grass or avenue trees, actually, these inner-city vegetations can be comprehensively captured because 1) we imported the vegetated training samples when identifying the impervious surfaces; 2) there are significant spectral and phenological differences between impervious surfaces and vegetations; 3) the high spatial resolution of Sentinel-1 and 2 can help us accurately identify these grass or avenue trees.

To intuitively understand the performance of GLC_FCS10 on the inner cities, Figure S3 gave the enlargements of three mega-cities in the Shanghai, Beijing and New York as examples, we can clearly find that these vegetations within the cities can be finely identified, e.g., street trees on both sides of the road and green belts in some neighborhoods.

[Figure]

Figure S3. The enlargements of three mega-cities in the Shanghai, Beijing and Paris, and the black circles illustrates the vegetations within the cities. The high-resolution imagery came from © Google Earth.

The problem of Imp-ESRI_LC also has been added in the manuscript as:

Beyond the confident impervious surface areas, it is equally important to identify high-quality natural land-cover types (Zhang et al., 2024a). To avoid confusion between natural land-cover types and impervious surfaces, the maximum impervious surface boundary (MaxBound_imp) is also generated. **The training samples for natural land-cover types should be located outside of the MaxBound_imp, i.e., some inner-city areas, easily misclassified or confused with impervious surfaces, will be excluded because Imp-ESRI_LC exhibits extensive patches of the impervious surface and lacks spatial details (Wang et al., 2024; Xu et al., 2024).**

2. Line 307, how many the samples for urban, rural and natural surfaces? The author just mentioned the ratio of these three land cover types.

Great thanks for the comment. The sample sizes of urban, rural and natural surfaces were selected as 5000, and the corresponding description has been added in the manuscript as:

Specifically, because we divide impervious surfaces training samples into rural and urban samples and design the equal distribution to enhance the training samples' ability to characterize impervious surfaces. The ratio of urban samples, rural samples, and natural surfaces is 1:1:1 for each $5 \times 5$ geographical tile. **Meanwhile, in terms of the sample size of each class, some previous studies have quantified the relationship between sample size and mapping accuracy (Foody, 2009; Li et al., 2014), and suggested a minimum size of 600 and maximum size of 8000 for these sparse and abundant land-cover types (Zhu et al., 2016). In this study, after considering the trade-off between sample representativeness with mapping efficiency, the sample size of each class was selected**

**as 5000, which was also consistent with the work of Zhang et al. (2022) in monitoring the impervious surface dynamics.**

3.Line 330, why divide the non-wetlands into water body, forest, grassland, bare land, and others but not the remaining 8 basic land cover types?

Great thanks for the comment. Yes, the non-wetlands should contain the remaining 8 basic land cover types, however, the snow and ice and shrubland usually belong to the sparse land-cover types at their co-existence areas, and some land-cover types share similar spectral characteristics are also merged into a common land-cover type. The specific descriptions about why we import non-wetland samples and

Wetlands are divided into four inland and three coastal wetland subcategories (in Table 2), and equal-distribution sampling is used to enhance the training samples' ability to characterize wetlands. **Additionally, since some non-wetland land-cover types also reflected the similar spectral characteristics with the wetlands, for example, the swamp and the forest/shrubland shared similar vegetation spectra during the peak growth period, while the marsh and cropland/grassland exhibited the characteristics of herbaceous vegetations, and the river flats also performed the spectral characteristics of bare land during the dry seasons (Zhang et al., 2023b).** Thus, the approximate ratio of inland wetlands, coastal wetlands, and non-wetlands (including **water body, forest/shrubland, cropland/grassland, bare land, and others**) is 4:3:5 in areas where they coexist.

4. Line 356 and Line 357, the manuscript references both "LCMAP_V" and "LCMAP_AL", are these two distinct datasets?

Great thanks for pointing out this mistake. Both of them belong to the same dataset, and it has been revised as "LCMAP_Val" in our revised manuscript.

5. Line 436, P.A. is complementary to the omission error, not the commission error.

Great thanks for pointing out this issue. It has been revised as

"Barren land has the lowest P.A. value of 31.93%, indicating a high omission error of 68.07%."

6. Why was the land cover type "Impervious surface" written as "Developed" in Table 5? These two have different definitions.

Great thanks for the comment. The reason why we use the "Developed" in the Table 5 because the third-party validation dataset (LCMAP_Val) contains the 634 **developed** points, and **one of the important guidelines for using the third-party validation points is not to artificially modify them**. To make the confusion matrix in Table 5 more intuitive, the column name of 'developed' has been changed as 'impervious' as:

| | Cropland | Forest | Grass/Shrub | Wetland | Impervious | Barren | Water | Ice & Snow | Total | U.A. |
|---|---|---|---|---|---|---|---|---|---|---|
| **Cropland** | 3445 | 28 | 393 | 6 | 0 | 9 | 2 | 0 | 3883 | 88.72 |
| **Forest** | 7 | 4621 | 133 | 92 | 0 | 0 | 2 | 0 | 4855 | 95.18 |
| **Grass/Shrub** | 368 | 358 | 3440 | 21 | 1 | 272 | 1 | 0 | 4461 | 77.11 |
| **Wetland** | 37 | 260 | 30 | 522 | 1 | 0 | 5 | 0 | 855 | 61.05 |
| **Developed** | 44 | 69 | 164 | 3 | 344 | 9 | 1 | 0 | 634 | 54.26 |
| **Barren** | 1 | 0 | 0 | 10 | 0 | 137 | 1 | 0 | 149 | 91.95 |
| **Water** | 0 | 2 | 1 | 63 | 2 | 1 | 1173 | 0 | 1242 | 94.44 |
| **Ice & Snow** | 0 | 0 | 0 | 0 | 0 | 1 | 0 | 2 | 3 | 66.67 |
| **Total** | 3902 | 5338 | 4161 | 717 | 348 | 429 | 1185 | 2 | 16082 | |
| **P.A.** | 88.29 | 86.57 | 82.67 | 72.80 | 98.85 | 31.93 | 98.99 | 100.00 | | |
| **O.A.** | | | | | 85.09 | | | | | |

| | |
|---|---|
| **Kappa** | 0.804 |

Meanwhile, the accuracy analysis about the different definition between 'developed' with 'impervious surface' was also discussed in our Discussion Section as:

**In addition, to objectively understand the accuracy performance of GLC_FCS10, we introduced the LCMAP_Val third-party validation dataset, but the differences in the definition of the classification system still affect the accuracy metrics, such as the higher P.A. and the lower U.A. for the impervious surfaces in Table 5. Therefore, one of the ongoing works would take some measures (such as: semantic similarity (Gao et al., 2020)) to more comprehensively and objectively assess the third-party accuracy metrics of GLC_FCS10.**

7. The confusion matrix (Table 5) shows that GLC_FCS10 misclassifies a large proportion of actual vegetation types as developed land (44 cropland, 69 forest and 164 grass/shrub reference samples are misclassified as developed land by GLC_FCS10), resulting in low UA for developed land, while only 4 developed land reference samples are misclassified elsewhere. This contradicts the statement in Line 433-435.

Great thanks for the comment. We are very sorry that the expression about the confusion matrix is not very clear, in fact, **the row direction represents the true value of LCMAP_Val and the column direction represents the predicted value of GLG_FCS10**. Namely, some vegetated developed pixels in the inner-city are excluded as impervious surfaces in the GLC_FCS10, so the GLC_FCS10 achieves the lowest U.A. of 54.26% with high P.A. of 98.85%.

**Response to comments**
**Paper #:** essd-2025-73
**Title:** GLC_FCS10: a global 10-m land-cover dataset with a fine classification system from Sentinel-1 and Sentinel-2 time-series data in Google Earth Engine
**Journal**: Earth System Science Data

**Reviewer #3**

The manuscript presents a new 10 m global land cover product, developed using a hierarchical methodology. This is a valuable and well-constructed contribution to global land cover mapping. The product is validated with over 56,000 samples and the LCMAP_Val dataset, which strengthens the credibility of the reported accuracy. However, several questions need further clarification to enhance the methodological rigor:

Great thanks for your positive comment, the manuscript has been further improved based on your and another reviewer's constructive and useful comments.

1.    Line 152: Why are impervious surfaces and wetlands treated separately from other land cover classes? While it is understandable that impervious surfaces are structurally different from natural land covers, the rationale for treating wetlands separately is less clear. Wetlands typically consist of a mix of vegetation and soil types, which may overlap with other land cover categories. Could this separation introduce additional uncertainty or classification confusion by inadvertently including other vegetation types? Please provide a clearer justification for this decision, especially compared to classifying all types together.

Great thanks for the comment. Yes, the reason why the impervious surface was treated separately from other land cover classes because impervious surfaces are structurally different from natural land covers.

In terms of wetlands, we have made wetlands independent for the following reasons: 1) a large amount of works have demonstrated that the spatial distributions of wetlands were simultaneously affected by the variations of water-levels and phenological information, and wetlands usually reflected more complicated spectral characteristics and spatiotemporal heterogeneities (Mao et al., 2020; Zhang et al., 2023;); 2) wetland distributions have strong locational characteristics (Gong et al., 2010), e.g., wetlands are mainly concentrated in low-lying areas and coastal wetlands are mainly distributed within 50 km of the coastal zone; 3) the existing land cover products, mixing wetlands with other natural surfaces, have been demonstrated to have poor performance in wetland mapping (Zhao et al., 2023).

Gong, P., Niu, Z., Cheng, X., Zhao, K., Zhou, D., Guo, J., Liang, L., Wang, X., Li, D., Huang, H., Wang, Y., Wang, K., Li, W., Wang, X., Ying, Q., Yang, Z., Ye, Y., Li, Z., Zhuang, D., Chi, Y., Zhou, H., and Yan, J.: China's wetland change (1990–2000) determined by remote sensing, Science China Earth Sciences, 53, 1036-1042, https://doi.org/10.1007/s11430-010-4002-3, 2010.

Mao, D., Wang, Z., Du, B., Li, L., Tian, Y., Jia, M., Zeng, Y., Song, K., Jiang, M., and Wang, Y.: National wetland mapping in China: A new product resulting from object-based and hierarchical classification of Landsat 8 OLI images, ISPRS Journal of Photogrammetry and Remote Sensing, 164, 11-25, https://doi.org/10.1016/j.isprsjprs.2020.03.020, 2020.

Zhang, X., Liu, L., Zhao, T., Chen, X., Lin, S., Wang, J., Mi, J., and Liu, W.: GWL_FCS30: a global 30 m wetland map with a fine classification system using multi-sourced and time-series remote sensing imagery in 2020, Earth Syst. Sci. Data, 15, 265-293, https://doi.org/10.5194/essd-15-265-2023, 2023.

Zhao, T., Zhang, X., Gao, Y., Mi, J., Liu, W., Wang, J., Jiang, M., and Liu, L.: Assessing the Accuracy and Consistency of Six Fine-Resolution Global Land Cover Products Using a Novel Stratified Random Sampling Validation Dataset, Remote Sensing, 15, 2285, https://doi.org/10.3390/rs15092285, 2023.

Based on the comment, the necessities of why impervious surfaces and wetlands are treated separately from other land cover classes have been added as:

To achieve high quality with detailed categorizations in global 10-m land-cover mapping, a hierarchical land-cover mapping methodology has been proposed. It leverages prior land-cover products and time-series satellite observations, **and gives more attention to impervious surfaces and wetlands by importing more prior knowledge and adding sufficient high-confidence training samples. Notably, the reasons why we separated impervious surfaces and wetlands from other land cover types are: 1) impervious surfaces are structurally different from natural land covers (Huang et al., 2022; Zhang et al., 2022); 2) wetland distributions have strong zonal characteristics (concentrating in low-lying areas) and reflect greatly complicated spectra and heterogeneities due to the variations of phenology and water-levels (Mao et al., 2020; Zhang et al., 2023b); 3) many previous studies have demonstrated that many existing global land-cover products suffered poor performance on these complicated land-cover types (Zhao et al., 2023).**

2.    Line 167: In the context of 10-meter resolution imagery, how are closed forests and open forests defined and differentiated?

Great thanks for the comment. The definitions of 30 fine land-cover types have been added in the Table S1 as:

**Table S1**. The detailed definitions of 30 land-cover types in the fine classification system.

| Fine classification system | Definition |
| --- | --- |
| Herbaceous rainfed cropland | Herbaceous cropland with no irrigation facilities and crops grown by natural precipitation |
| Tree or shrub covered rainfed cropland (orchard, oil palm…) | Tree or shrub covered rainfed cropland, mainly including orchard, oil palm, etc. |
| Irrigated cropland | Cropland with guaranteed water sources and irrigation facilities that can be irrigated normally in a typical year |
| Closed evergreen broadleaved forest | Evergreen broadleaved tree cover, tree height > 3 m, tree-cover percentage > 40% |
| Open evergreen broadleaved forest | Evergreen broadleaved tree cover, tree height > 3 m, 15%< tree-cover percentage < 40% |
| Closed deciduous broadleaved forest | Deciduous broadleaved tree cover, tree height > 3 m, tree-cover percentage > 40% |
| Open deciduous broadleaved forest | Deciduous broadleaved tree cover, tree height > 3 m, 15%< tree-cover percentage < 40% |
| Closed evergreen needleleaved forest | Evergreen needleleaved tree cover, tree height > 3 m, tree-cover percentage > 40% |
| Open evergreen needleleaved forest | Evergreen needleleaved tree cover, tree height > 3 m, 15%< tree-cover percentage < 40% |
| Closed deciduous needleleaved forest | Deciduous needleleaved tree cover, tree height > 3 m, tree-cover percentage > 40% |
| Open deciduous needleleaved forest | Deciduous needleleaved tree cover, tree height > 3 m, 15%< tree-cover percentage < 40% |
| Closed mixed-leaf forest | Mixed broadleaved and needleleaved forests, tree height > 3 m, tree-cover percentage > 40% |
| Open mixed-leaf forest | Mixed broadleaved and needleleaved forests, tree height > 3 m, 15%< tree-cover percentage < 40% |
| Evergreen shrubland | Vegetation communities dominated by low cover and evergreen dwarf and scrub woodlands |
| Deciduous shrubland | Vegetation communities dominated by woody shrubs that lose their leaves in winter or the dry season |
| Grassland | Refers to land where herbaceous plants predominate |
| Lichens and mosses | Lichens and moss-covered areas |
| Swamp | The forest or shrubs which grow in the inland freshwater |
| Marsh | Herbaceous vegetation (grasses, herbs and low shrubs) grows in the freshwater |
| Lake/river flat | The non-vegetated flooded areas along the rivers and lakes |
| Saline | Characterized by saline soils and halophytic (salt tolerant) plant species along saline lakes |
| Mangrove forest | The forest or shrubs which grow in the coastal brackish or saline water |
| Salt marsh | Herbaceous vegetation (grasses, herbs and low shrubs) in the upper coastal intertidal zone |
| Tidal flat | The tidal flooded zones between the coastal high and low tide levels including mudflats and sandflats |
| Urban impervious surfaces | Land covered with buildings and other man-made structures within the urban boundary |
| Rural impervious surfaces | Land covered with man-made structures outside the urban boundary, mainly including rural residential land, transportation land, etc. |
| Sparse vegetation | Areas covered by woodland, shrubs and grasses, vegetation-cover percentage < 15% |
| Bare areas | Refers to land that is largely devoid of vegetation cover |
| Water | Lakes, rivers and streams that are always flooded |
| Permanent ice and snow | Areas covered by snow and ice all year round |

Given the spatial resolution, the criteria used to distinguish these two types may significantly influence classification reliability.

In this study, the criteria used to distinguish the open forest and closed forest is the tree-cover percentage. And we agree that there may be significant mixing problems for forested areas with moderate tree cover. Thus, one of our ongoing works is retrieving the fraction of tree-cover (FTC), and then import the annual maximum FTC to better distinguish the open forest and closed forest.

3. Line 191: Regarding the use of MaxBound, does it only constrain the region from which training samples are selected, or does it also limit the area where land cover classification is applied? Please clarify its role in both processes.

Great thanks for the comment. Yes, the MaxBound_imp was not only use to generate the training samples but also used in subsequent classification. The role of MaxBound_imp has been added in the subsequent classification as (Section 3.4.1):

When building the training model for each $5 \times 5$ geographical tile, we also import training samples within their spatial neighborhood of $3 \times 3$ tiles to ensure spatial consistency over the adjacent tiles. **Since the $MaxBound_{imp}$ (Eq. (2)) provides the maximum potential areas of impervious surfaces because of the overestimation problem of Imp-ESRI_LC (Wang et al., 2024; Xu et al., 2024), all identified impervious surfaces should be within the $MaxBound_{imp}$.** Afterward, we can produce 984 $5 \times 5$ impervious surface and natural land cover maps using the local adaptive modeling strategy.

4. Lines 277 and 289: It appears that you apply a percentile-based compositing method to mitigate the impact of clouds and shadows, thereby avoiding the direct use of full time series satellite data. However, this method does not fully eliminate inter-tile differences, particularly in areas with high cloud frequency. Consequently, mosaic seams may still be visible. Could the authors clarify whether any additional techniques were applied to address these residual seams?

Great thanks for the comment. Yes, we completely agree that the percentile-based method does not fully eliminate inter-tile differences, particularly in areas with high cloud frequency. In terms of our land-cover classifications, these influences are minimized through the integration of Sentinel-1 and Sentinel-2 imagery and the import of spatial textures from the gray level co-occurrence matrix.

However, as for these persistent-cloudy areas, the lack of efficient optical observations still affects the continuity of land-cover maps, and causes the mosaic seams. Recently, some previous works have explained that the harmonization of Landsat and Sentinel-2 can increase the likelihood of clear observations, and some researches used the deep learning models to improve the land-cover mapping performance on these cloudy areas. Thus, this limitation is also discussed on the Section 4.5 as:

**Meanwhile, although the combination of time-series Sentinel-1 and Sentinel-2 can minimize the effect of clouds and shadows, some high cloud-contaminated areas might be still affected by these persistent clouds, i.e., mosaic seams may be visible in these special areas. Many previous studies have demonstrated that the harmonization of Landsat and Sentinel-2 can increase the likelihood of clear observations (Claverie et al., 2018), and the advances of deep learning models also improve the land-cover mapping performance on these cloudy areas (Xu et al., 2024a). Thus, how to combine the Landsat imagery and deep learning techniques to further improve the quality of GLC_FCS10 in the persistent cloudy areas will be one of the future works.**

5. Line 345: What happens if a specific land cover type is absent from the training samples within a 3×3 tile region due to limited sample size? Would this omission lead to that type being entirely excluded from prediction in the region? If so, how does the method ensure completeness of classification in regions with rare or underrepresented classes?

Great thanks for the comment. Actually, we have built a backup global training sample library, storing the typical training features of all land-cover types over the globe, to avoid missing training samples of these sparse or underrepresented land-cover types.

However, after using the training samples from neighboring 3 × 3 geographical tiles, the missing training samples in the central tile almost were supplemented by neighboring 3 × 3 tiles, which caused the backup library to lose its function.

6.    Line 408: Why is the confusion matrix reported for only 16 land cover types, while the final product contains 30 classes? Please explain the rationale behind this evaluation subset and whether accuracy metrics for the remaining classes are available.

Great thanks for the comment. This is a good question, the reason why we only give the confusion matrix for 16 land-cover types because of **the limitations of the global validation dataset**. If the validation points are further refined, we cannot guarantee the authenticity and reliability of the validation points through visual interpretation and auxiliary information. Thus, this limitation has been discussed on the Section 4.5 as:

[revised manuscript text omitted]

---

## Author Response (AR2)

Dear Topical Editor and Reviewers:

On behalf of my co-authors, we thank you very much for reviewing our manuscript and giving us a lot of useful comments and suggestions. We appreciate the comments on our manuscript entitled "*GLC_FCS10: a global 10-m land-cover dataset with a fine classification system from Sentinel-1 and Sentinel-2 time-series data in Google Earth Engine*" (essd-2025-73).

We have revised the manuscript carefully according to the comments. All the changes were high-lighted (red color) in the manuscript. And the point-by-point response to the comments of the reviewers is also listed below.

Looking forward to hearing from you soon.

Best regards,

Prof. Liangyun Liu liuly@radi.ac.cn

Institute of Remote Sensing and Digital Earth, Chinese Academy of Sciences

No.9 Dengzhuang South Road, Haidian District, Beijing 100094, China

**Response to comments**
**Paper #:** essd-2025-73
**Title:** GLC_FCS10: a global 10-m land-cover dataset with a fine classification system from Sentinel-1 and Sentinel-2 time-series data in Google Earth Engine
**Journal**: Earth System Science Data

We have received comments from reviewers regarding your manuscript. The review outcome is positive and I am pleased to inform you that, in principle, the manuscript will be accepted. But before the formal publication, please check the language thoroughly, particularly the newly added text. Some statements with grammar mistakes or confusion are listed below:

Great thanks for the comments. Based on your suggestions, the manuscript has been carefully checked, and the grammar mistakes or confusions have been clarified or corrected.

L161: many previous studies have demonstrated that many existing global land-cover products suffered poor performance on these complicated land-cover types.

Great thanks for the comment. Yes, that's a confusing sentence to read, thus, the statement has been revised as:

**some previous studies emphasized that taking additional measures (such as: importing more prior knowledge, thematic mapping and so on) was an effective means of improving the accuracy of wetlands and impervious surfaces (Gong et al., 2020; Zhang et al., 2023b).**

L246-247: Fortunately, the ongoing updating of GLC_FCS30D is still in progress, the land-cover change masks during 2022-2023 have been finished, and which are also used to remove these changed and low-confidence areas.

Great thanks for the comment. It has been revised as:

Fortunately, the ongoing updating of GLC_FCS30D is still in progress, the land-cover change masks during 2022-2023 have been finished, **and which are used to guarantee the temporal consistency between prior land-cover products with the training areas.**

[revised manuscript text omitted]